# Comparison of fecal and blood metabolome reveals inconsistent associations of the gut microbiota with cardiometabolic diseases

Kui Deng [1,2,3,4,5], Jin-jian Xu[1,5], Luqi Shen[2,3,4,5], Hui Zhao[2,3,4], Wanglong Gou[2,3,4], Fengzhe Xu[2,3,4], Yuanqing Fu [2,3,4], Zengliang Jiang[2,3,4], Menglei Shuai[2,3,4], Bang-yan Li[1], Wei Hu[1], Ju-Sheng Zheng [2,3,4] ✉ & Yu-ming Chen [1] ✉

Blood metabolome is commonly used in human studies to explore the associations of gut microbiota-derived metabolites with cardiometabolic diseases. Here, in a cohort of 1007 middle-aged and elderly adults with matched fecal metagenomic (149 species and 214 pathways) and paired fecal and blood targeted metabolomics data (132 metabolites), we find disparate associations with taxonomic composition and microbial pathways when using fecal or blood metabolites. For example, we observe that fecal, but not blood butyric acid significantly associates with both gut microbiota and prevalent type 2 diabetes. These findings are replicated in an independent validation cohort involving 103 adults. Our results suggest that caution should be taken when inferring microbiome-cardiometabolic disease associations from either blood or fecal metabolome data.

The human gut microbiome is a diverse and complex ecosystem, with a huge number of microbes inhabited in the gastrointestinal tract. Gut microbes play essential roles in maintaining human health, and can regulate the host physiology and disease risk through producing functional metabolites such as short-chain fatty acids (SCFAs) and bile acids[1]. These metabolites are either derived directly from microbes, from exogenous dietary residue or endogenous substrates generated by the host[2,3]. Gut microbiota-related metabolites serve as important intermediates in the cross-talk between the gut microbiota and host, substantially influencing the development of cardiometabolic diseases[1,3,4].

Feces and blood are both commonly used biological samples to capture the gut microbial metabolites. Previous evidence demonstrated that gut microbiota significantly explained the variance of blood metabolome (average explained variance: 4.6–15%)[5–8], and meanwhile, other studies showed that the gut microbial composition was largely reflected by fecal metabolome (average explained variance: 67.7%)[9]. In addition, diet and genetics are also important factors shaping blood and fecal metabolome[5,6,8–11]. In epidemiological studies, blood was the widely used biological sample for metabolome profiling to facilitate the exploration of the association between gut microbiota-related metabolites and cardiometabolic diseases[12–17]. For example, Nemet et al. found that blood phenylacetylglutamine was positively associated with incident cardiovascular disease and major adverse cardiovascular events[14]. Vangipurapu et al. identified several gut microbiota-related blood metabolites including kynurenate, dimethylglycine, 2-hydroxyhippurate that were associated with the risk of type 2 diabetes (T2D)[12]. Circulating trimethylamine N-oxide was associated with cardiovascular disease risk[17].

Currently, the majority of gut microbiota-metabolome association studies were performed using either feces or blood for metabolome profiling[5,6,9,18,19]. Although both fecal and blood metabolomics data were used in several prior studies to explore the gut microbiota-metabolome associations, fecal and blood

[1]Guangdong Provincial Key Laboratory of Food, Nutrition and Health, Department of Epidemiology, School of Public Health, Sun Yat-sen University, Guangzhou 510080, China. [2]Westlake Intelligent Biomarker Discovery Lab, Westlake Laboratory of Life Sciences and Biomedicine, Hangzhou 310024, China. [3]Key Laboratory of Growth Regulation and Translational Research of Zhejiang Province, School of Life Sciences, Westlake University, Hangzhou 310024, China. [4]Institute of Basic Medical Sciences, Westlake Institute for Advanced Study, Hangzhou 310024, China. [5]These authors contributed equally: Kui Deng, Jin-jian Xu, Luqi Shen. ✉e-mail: zhengjusheng@westlake.edu.cn; chenyum@mail.sysu.edu.cn

samples were not collected at the same time points and were obtained from different subsets of participants[20], or the sample size was very small ($N < 100$)[10,21]. Therefore, direct comparison between paired fecal and blood metabolome in their associations with gut microbiota and cardiometabolic diseases has been rare but highly warranted, as it could help guide the practical applications of fecal and blood metabolomics technology in studying the associations of gut microbiota with cardiometabolic diseases in human studies.

To fill the above research gaps, we systematically investigated the associations of gut microbiota with paired fecal and blood metabolites in 1007 middle-aged and elderly Chinese adults, with multi-omics datasets of gut metagenomic sequencing (using fecal samples), and targeted quantitative fecal and blood metabolome collected at the same time point. We identified several fecal and blood metabolites that were well-predicted by the gut microbiota, and then assessed their associations with cardiometabolic diseases, including T2D, obesity, nonalcoholic fatty liver disease (NAFLD), and hypertension.

## Results
### Study overview
This study was based on the Guangzhou Nutrition and Health Study (GNHS)[22,23], in which 1007 participants (age: $64.7 \pm 5.6$) whose stool and blood samples were collected at the same day, without taking any antibiotics within two weeks, were included in our present study as a discovery cohort. We included 103 participants (age: $71.5 \pm 7.1$) from the control arm of a hip fraction case-control study with paired fecal and blood samples as an external validation cohort[24].

We used the shotgun metagenomic sequencing and targeted quantitative fecal and serum metabolomics profiling to generate metagenome, and fecal and blood metabolome data, respectively, for both discovery and validation cohort. After excluding metabolites with over 20% missing values or with relative standard deviation (standard deviation/mean) $\geq 0.3$ in quality control samples, 132 matched fecal and blood metabolites were remained for subsequent analysis. These metabolites mainly consisted of known gut microbiota-derived metabolites including SCFAs, bile acids, indoles, etc. and other key host metabolites, such as amino acids, carbohydrates, organic acids, and so on. These metabolites covered a wide range of metabolic classes, including amino acids, fatty acids, organic acids, carbohydrates, bile acids, benzenoids, carnitines, phenylpropanoic acids, pyridines, indoles, organooxygen compounds, and nucleosides. The overview of this study is showed in Fig. 1.

### Correlations between paired fecal and blood metabolites
We first assessed the phenotypic correlations (a direct comparison of each metabolite's values between feces and blood) between paired fecal and blood metabolites using partial *Spearman* correlation adjusted for age, sex, and BMI. The correlations between the paired fecal and blood metabolites were generally low (phenotypic correlation coefficients [mean ± sd]: $0.05 \pm 0.12$). There were only eight significantly correlated metabolites between feces and blood (correlation coefficients > 0.3 and FDR < 0.05), including 2-phenylpropionate, 3-indolepropionic acid, eicosapentaenoic acid (EPA), glycodeoxycholic acid, glycoursodeoxycholic acid, hydrocinnamic acid, N-Methylnicotinamide, and phenylacetic acid (Fig. 2a, Supplementary Fig. 2 and Supplementary Data 1). Partial *Spearman* correlation analysis only adjusted for age and sex showed consistent results ($r = 0.999$, $P < 0.0001$; Supplementary Fig. 3).

We then examined the genetic correlations (the proportion of shared heritability between paired fecal and blood metabolites) between the paired fecal and blood metabolites using bivariate GREML analysis[25]. Consistent with the results of phenotypic correlations, there were low genetic correlations among most paired fecal and blood metabolites (genetic correlation coefficients [mean ± sd]: $0.13 \pm 0.75$),

with only 16 metabolites having an absolute value of genetic correlation coefficients > 0.3 and FDR < 0.05 (Fig. 2a, Supplementary Fig. 2 and Supplementary Data 1). Among them, three metabolites (3-indolepropionic acid, glycoursodeoxycholic acid, and phenylacetic acid) overlapped with those of phenotypic correlations (Fig. 2b).

### Comparison between paired fecal and blood metabolome in their associations with taxonomic composition and microbial pathways
Given that there were low correlations between most paired fecal and blood metabolites, we explored whether there were differences between the associations of taxonomic composition or microbial pathways with paired fecal and blood metabolites. We used two machine learning models, random forest (RF) and Light Gradient Boosting Machine (LightGBM) with five-fold cross-validation to independently examine the association between taxonomic composition/microbial pathways and each metabolite, either in the fecal or blood samples, and then compared whether the associations were consistent across feces and blood. As the RF model had a better performance than the LightGBM model with a smaller root mean square error and more well-predicted metabolites (Supplementary Fig. 1, Supplementary Notes 1), the main results of this study were reported based on the RF model. The predictability of fecal metabolites based on taxonomic composition or microbial pathways, which was measured by *Spearman*'s correlation between the measured and predicted metabolite levels for held-out samples by machine learning pipeline, was much higher than those of blood metabolites (Fig. 3a, b, Supplementary Fig. 4a and Supplementary Fig. 4b, Supplementary Data 2). Overall, compared with blood metabolites, fecal metabolites had significantly higher correlations with taxonomic composition (correlation coefficients: mean±sd: $0.40 \pm 0.15$ vs. $0.09 \pm 0.11$; $P < 0.0001$ by Wilcoxon signed-rank test; Fig. 3c) and microbial pathways (correlation coefficients: mean±sd: $0.32 \pm 0.14$ vs. $0.06 \pm 0.11$; $P < 0.0001$ by Wilcoxon signed-rank test; Supplementary Fig. 4c).

We then considered metabolites with correlation coefficient > 0.3 and FDR < 0.05 as well-predicted metabolites[18,26]. Based on taxonomic composition, we identified 98 well-predicted fecal metabolites, including several known gut microbiota-derived metabolites, such as SCFAs (acetic acid, butyric acid), bile acids (glycoursodeoxycholic acid, tauroursodeoxycholic acid, glycocholic acid, taurochenodeoxycholic acid, glycodeoxycholic acid, glycochenodeoxycholic acid), phenylpropanoic acids (2-phenylpropionate, hydrocinnamic acid, hydroxyphenyllactic acid, 3-3-Hydroxyphenyl-3-hydroxypropanoic acid), benzenoids (phenylacetic acid, 4-hydroxyphenylpyruvic acid, phenylpyruvic acid), indoles (3-indolepropionic acid), and so on (Fig. 3a, e and Supplementary Data 2). There were 10 well-predicted blood metabolites (Fig. 3b, e and Supplementary Data 2), among which, 8 overlapped with fecal samples (2-phenylpropionate, 3-indolepropionic acid, glutaric acid, glycodeoxycholic acid, glycoursodeoxycholic acid, hydrocinnamic acid, N-Methylnicotinamide, and phenylacetic acid). There were 90 metabolites that were only well-predicted in feces and not in blood, and 2 metabolites (alpha-N-Phenylacetyl-L-glutamine and hippuric acid) that were only well-predicted in blood and not in feces. Most well-predicted fecal metabolites had superior predictability over their corresponding paired blood metabolites, and for metabolites that were well-predicted in both feces and blood, the models of most well-predicted fecal metabolites significantly outperformed their respective models of well-predicted blood metabolites (FDR < 0.05, see Methods; Fig. 3d, Supplementary Fig. 7a and Supplementary Data 3). The identified well-predicted metabolites based on microbial pathways showed consistent results, in which all identified well-predicted metabolites based on microbial pathways belonged to the identified well-predicted metabolites based on taxonomic composition (Supplementary Fig. 4, Supplementary Fig. 5a and Supplementary Fig. 5b and Supplementary Fig. 7b;

**Fig. 1 | Workflow of the present study.** A total of 1007 participants from the Guangzhou Nutrition and Health Study with matched gut metagenomic and fecal and blood metabolomics data and without taking antibiotics within two weeks are included in this study. Shotgun metagenomic sequencing is performed for fecal samples to obtain the metagenomic data, including taxonomic composition and microbial pathways. A targeted metabolome profiling is performed to obtain the fecal and blood metabolomics data. After removing metabolites with missing rate ≥ 0.2 or with relative standard deviation ≥ 0.3, 132 matched fecal and blood metabolites are remained for subsequent analysis. We estimate the gut microbiota-fecal/blood metabolite associations using the machine learning pipeline, and compare the associations of taxonomic composition/microbial pathways with paired fecal and blood metabolites. We then explore the associations of gut microbiota-related fecal and blood metabolites with cardiometabolic diseases. We further replicate the identified significant associations in an independent validation cohort. This figure has been designed using images from Flaticon.com. NAFLD nonalcoholic fatty liver disease.

Supplementary Data 2–3). For sensitivity analysis with different cut-off $r$ values (correlation coefficient: 0.2 and 0.4), the number of well-predicted fecal metabolites was substantially more than that of blood metabolites (Supplementary Fig. 5c, d). Sensitivity analysis for gut microbiota-fecal/blood metabolite associations among participants without T2D, hypertension or dyslipidemia medications showed consistent results (Supplementary Fig. 6).

We then validated the robustness of the above associations between taxonomic composition/microbial pathways and well-predicted metabolites in an independent validation cohort.

We trained RF models for each well-predicted metabolite based on taxonomic composition or microbial pathways in the discovery cohort, and directly applied them to the validation cohort. Based on *Spearman*'s correlation coefficient > 0.3 and FDR < 0.05 (Methods), 68% (132/194) associations could be validated, including 63.3% (62/98) taxonomic composition-fecal metabolite associations, 80% (8/10) taxonomic composition-blood metabolite associations, 71.4% (55/77) microbial pathways-fecal metabolite associations, and 77.8% (7/9) microbial pathways-blood metabolite associations (Fig. 3e, Supplementary Fig. 4e and Supplementary Fig. 9 and Supplementary Data 4).

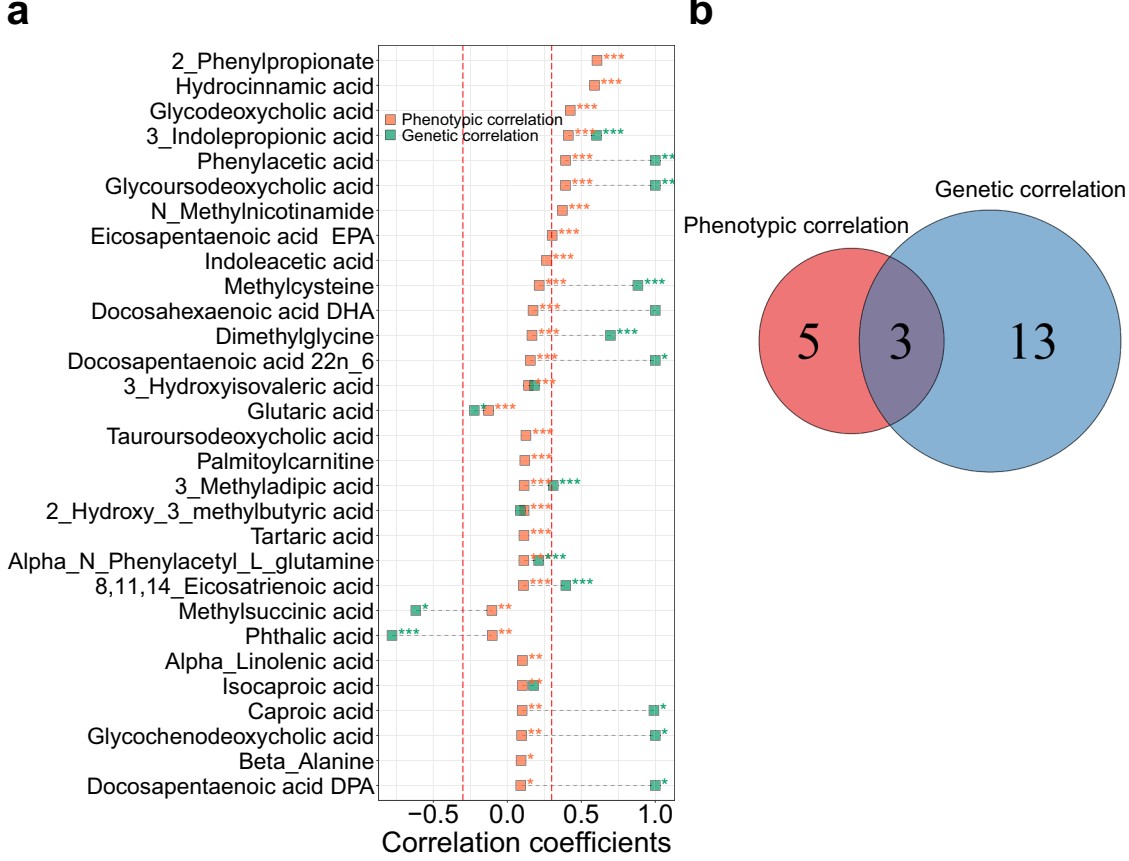

**Fig. 2 | Phenotypic and genetic correlations between paired fecal and blood metabolites. a** Phenotypic and genetic correlations between paired fecal and blood metabolites for the top 30 metabolites that are ranked by phenotypic correlations. Phenotypic correlations between paired fecal and blood metabolites are estimated by partial *Spearman* correlation analysis, adjusted by age, sex and BMI. Genetic correlations are calculated using bivariate GREML analysis. Correlations with FDR < 0.05 and |*r*| > 0.3 (red dashed lines) are considered significant. FDR is controlled by the Benjamini-Hochberg method. \*FDR < 0.05, \*\*FDR < 0.01, \*\*\* FDR < 0.005. The results for remaining metabolites are shown in Supplementary Fig. 2. For genetic correlation analysis, there are 38 metabolites that do not converged during the calculation process and thus have no results. **b** The overlap between significant phenotypic and genetic correlations. All statistical tests are two-sided. Source data are provided as a Source Data file.

## Associations of well-predicted metabolites with cardiometabolic diseases

Subsequently, we explored the associations of the above well-predicted fecal and blood metabolites identified in the discovery cohort with prevalent cardiometabolic diseases (T2D, obesity, NAFLD, and hypertension). There were 12 well-predicted fecal metabolites being associated with several cardiometabolic diseases including T2D (7 significant associations), obesity (4 significant associations) and NAFLD (1 significant association) (FDR < 0.05 by the multivariable logistic models; Fig. 4a and Supplementary Data 5). Sensitivity analysis with an additional adjustment of medications for T2D, hypertension, and dyslipidemia showed consistent results (*r* = 0.998, *P* < 0.0001; Supplementary Fig. 8). However, we did not find any significant association for the well-predicted blood metabolites (Fig. 4b and Supplementary Data 5).

These findings were interesting for metabolites that were well-predicted in both blood and feces. For example, there were nominal associations of blood 2-phenylpropionate, 3-indolepropionic acid and hydrocinnamic acid with T2D (*P* < 0.05 and FDR > 0.05), while the associations of fecal levels of these metabolites with T2D were statistically significant (FDR < 0.05). We further validated the significant associations of these fecal metabolites with cardiometabolic diseases in the external validation cohort. The low replication rate shown in Fig. 4c was largely a power issue as the sample size of the validation cohort was small. However, we discovered that 8 out of 11 associations for the well-predicted fecal metabolites had the same effect directions and strong correlation between partial regression coefficients obtained from the discovery and validation cohort (*r* = 0.894; Fig. 4c, d), which suggested that the majority of our identified associations between well-predicted fecal metabolites and cardiometabolic diseases were consistent across the discovery and validation cohort.

## Comparison between paired fecal and blood SCFAs in associating gut microbiota and T2D

SCFAs, such as acetic acid, propionic acid, and butyric acid are well-known gut microbiota-derived metabolites that are produced by microbial fermentation of dietary fiber[1]. Both fecal and blood levels of SCFAs have been commonly used to explore the role of SCFAs in human health[27–31]. Through comparing paired fecal and blood SCFAs in their associations with taxonomic composition, we found that gut microbes were strongly associated with fecal levels of acetic acid (*r* = 0.53, FDR < 0.0001; Fig. 5a) and butyric acid (*r* = 0.59, FDR < 0.0001; Fig. 5c), but were not associated with blood acetic acid (*r* = 0.02, FDR = 0.6797; Fig. 5b) or butyric acid (*r* = 0.03, FDR = 0.3605; Fig. 5d). Similar results were observed for the associations of microbial pathways with paired fecal and blood SCFAs (Supplementary Fig. 10). Furthermore, we found that fecal butyric acid levels were inversely associated with prevalent T2D (*β* = −0.25, FDR = 0.0375; Fig. 5e), while blood butyric acid was not associated (*β* = −0.05, *P* = 0.543; Fig. 5f).

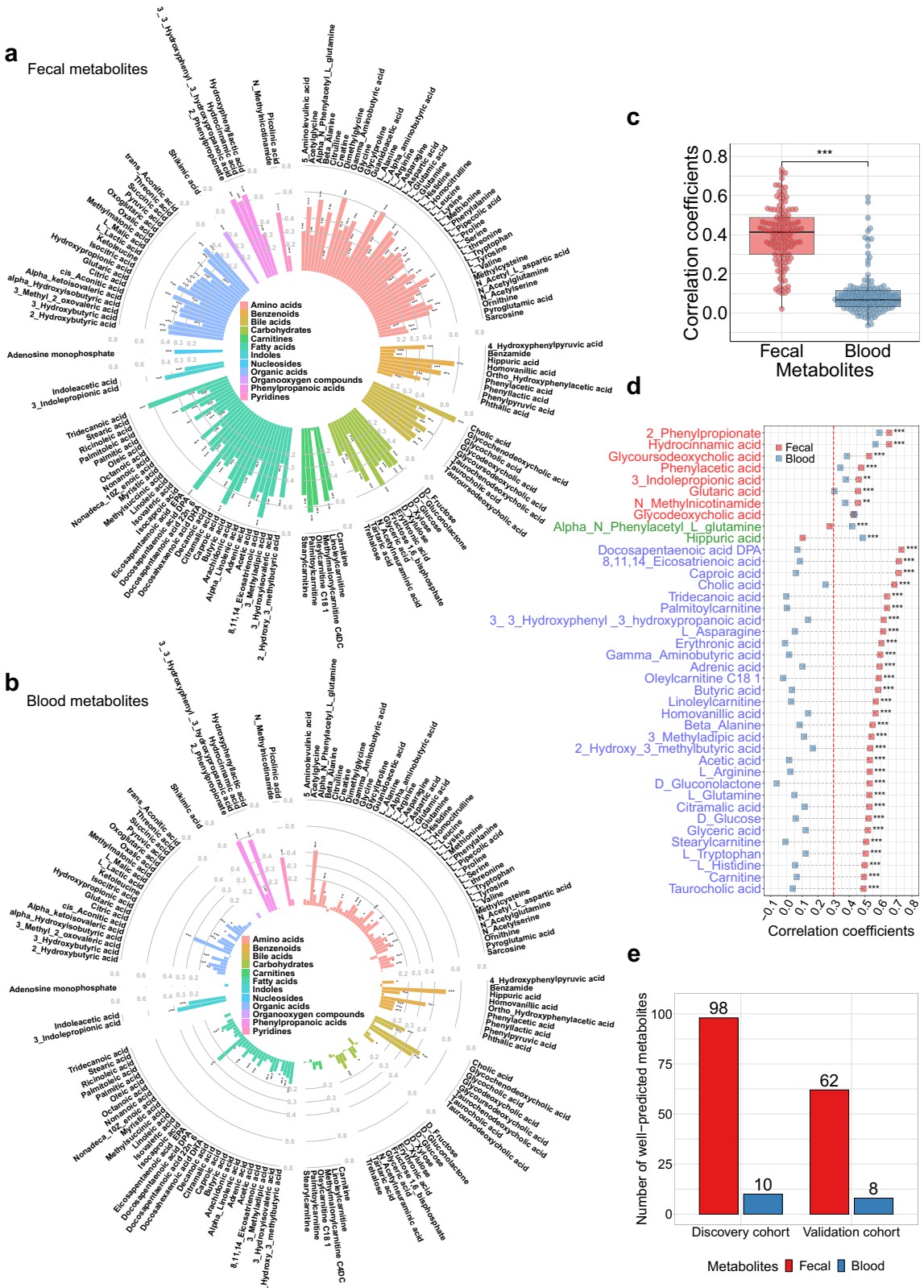

## Discussion

In the present population-based large-scale gut microbiota-metabolome association study, we compared paired fecal and blood metabolome in their associations with gut microbiota and cardiometabolic diseases. We found that taxonomic composition and microbial pathways were more broadly associated with fecal metabolites compared with blood metabolites. We further showed that some blood metabolites were weakly associated with gut microbiota or cardiometabolic diseases. These results suggested that caution should be taken when inferring microbiome-cardiometabolic disease associations from either blood or fecal metabolome data.

The importance of fecal metabolome for understanding the gut microbiota function in humans has been recognized in the past few years[9]. Meanwhile, blood metabolome was also closely linked with gut

**Fig. 3 | Comparisons between paired fecal and blood metabolites in their associations with taxonomic composition. a** The associations of taxonomic composition with fecal metabolites, and **b** with blood metabolites. The random forest model with five-fold cross-validation is used to predict the fecal or blood metabolite levels based on taxonomic composition. *Spearman*'s correlation between measured and predicted metabolite levels is used to measure the association of taxonomic composition with fecal or blood metabolites. *FDR < 0.05, **FDR < 0.01, *** FDR < 0.005. **c** The distributions of the associations of taxonomic composition with fecal and blood metabolites (*n* = 132). The difference between the distributions of taxonomic composition-fecal metabolite associations and taxonomic composition-blood metabolite associations is tested by Wilcoxon signed-rank test. Box plots indicate median and interquartile range (IQR). The upper and lower whiskers indicate 1.5 times the IQR from above the upper quartile and below the lower quartile. ***P < 0.0001. **d** Differences between the associations of taxonomic composition with paired fecal and blood metabolites that are well-predicted in both feces and blood (marked with red in y axis), metabolites that are only well-predicted in blood and not in feces (marked with green in y axis), and the top 30 metabolites that are only well-predicted in feces and not in blood (marked with blue in y axis). Metabolites are ranked by the predictability of fecal metabolites. Differences between the associations of taxonomic composition/microbial pathways with paired fecal and blood metabolites are tested by the method proposed by Hittner et al. (see "Methods"). *FDR < 0.05, **FDR < 0.01, ***FDR < 0.005. The results for the top 31–90 metabolites that are only well-predicted in feces and not in blood are presented in Supplementary Fig. 7a. **e** The number of well-predicted fecal and blood metabolites based on taxonomic composition and the number of validated associations between taxonomic composition and well-predicted fecal/blood metabolites in the validation cohort. Well-predicted metabolites are defined as *Spearman*'s correlation coefficient > 0.3 and FDR < 0.05. FDR is controlled by the Benjamini–Hochberg method. Associations with *Spearman*'s correlation coefficient > 0.3 and FDR < 0.05 are considered as being validated in the validation cohort. All statistical tests are two-sided. Source data are provided as a Source Data file.

microbiota in several recent studies[5–8]. Although paired fecal and blood metabolome data are collected in some previous studies[10,21], their samples sizes were very small (*N* < 100) and direct comparisons between paired fecal and blood metabolites in their associations with gut microbiota were not performed. To the best of our knowledge, our study was the largest study comparing simultaneously collected paired fecal and blood metabolome for their associations with gut microbiota, which could potentially provide the guidance for the practical applications of fecal and blood metabolome in exploring the associations of gut microbiota-related metabolites with host health.

Human blood is a conventionally used biological sample for metabolome profiling in epidemiological studies, where the blood metabolome was used to explore the associations of gut microbiota-related metabolites with cardiometabolic diseases[12–17]. Metabolites that are produced by the gut bacteria may experience complex intermediate processes, such as intestinal epithelial absorption, enterohepatic circulation, and liver absorption and transformation, when transporting from intestinal tract to the bloodstream[1,32], which may weaken the associations between gut microbiota and blood metabolites. In addition, when gut metabolites enter the blood, they may be maintained within physiological ranges by homeostasis that is achieved substantially through mass action-driven oxidation[33]. Thus, blood metabolites may be more stable than fecal metabolites and may be less sensitive to the change of gut microbiota.

Our study identified several known gut microbiota-derived metabolites that were associated with cardiometabolic diseases. Butyric acid was one of the three main SCFAs (acetic acid, propionic acid, and butyric acid) derived by gut microbiota in human gut. The fecal butyric acid levels were inversely (beneficially) associated with T2D in our present study, which has been demonstrated by several animal models[34,35]. In human studies, higher fecal butyric acid levels were associated with improved insulin response[28]. Ana et al. enrolled 20 adults (9 had T2D) aged ≥50 years and found that the concentration of fecal butyric acid was lower in T2D participants compared to those without T2D[36]. We found that fecal hydrocinnamic acid levels were inversely associated with T2D, which was consistent with Vangipurapu et al.'s study, where blood levels of this metabolite were measured[12]. Our study found that there was an inverse association between fecal 2-phenylpropionate and T2D. The physiological function of 2-phenylpropionate was rarely explored in the literature, and its underlying mechanism involved in T2D development should be further clarified. Interestingly, the concentration of blood 2-phenylpropionate, which was also associated with gut microbiota, was not associated with T2D. This further supported the strength of fecal metabolome in associating gut microbiota and metabolic diseases. However, capturing associations is just the first step in epidemiological studies aiming at providing the directions for downstream mechanistic investigations as how the gut microbiota-derived metabolites affect cardiometabolic diseases, in which blood metabolome may play an important role.

Our study has several strengths. First, this study has a large sample size with matched gut metagenomic, and fecal and blood metabolomics data collected simultaneously within each participant, which facilitates paired fecal and blood metabolome comparison. Second, targeted fecal and blood metabolome profiling is performed in our study participants, which provides absolute quantitative metabolomics data. Third, most of our findings could be validated in an independent validation cohort.

There are also several limitations. First, associations identified in this study are not necessarily causal. Specifically, those identified well-predicted metabolites based on gut microbiota are not necessarily derived from gut microbiota, although there are high correlations. Second, there are only 132 metabolites analyzed in our study because of the targeted metabolome profiling. These metabolites constitute only a small fraction of metabolic spectrum. Therefore, our results should be validated in large cohorts with a large number of measured metabolites in the future. Third, our cohort only surveyed the antibiotics use within 2 weeks at the time of stool sample collection. As the effects of antibiotic treatments can last longer time, our findings should be further validated in other large cohorts that surveyed the long-term antibiotics use (>6 months). Fourth, there are only 103 participants in the validation cohort with the number of patients 27, 23, and 14 for T2D, hypertension, and obesity, respectively, which limits the statistical power to fully replicate the results of metabolite-cardiometabolic disease associations. In addition, as NAFLD status is not available in the validation cohort, the corresponding associations are unable to be validated. Thus, larger replication cohorts are needed in future to further replicate our findings. Finally, as the participants enrolled in our study are not selected at random, there is potential selection bias. Additionally, as our cohorts are only based on individuals of Chinese ethnicity, the generalization of our conclusions to populations with other ethnicities needs to be further investigated.

In conclusion, we conducted a comparison between paired fecal and blood metabolome in their associations with gut microbiota and cardiometabolic diseases in a large cohort study. Our results showed disparate associations with gut microbiota and cardiometabolic diseases when using fecal or blood metabolites. In epidemiological studies, caution should be taken when inferring microbiome-cardiometabolic disease associations from either blood or fecal metabolome data.

## Methods
### Study cohorts
This study was based on the Guangzhou Nutrition and Health Study (GNHS). The detailed description of GNHS cohort could be found in previous studies[22,23]. Briefly, a total of 4048 Chinese participants aged

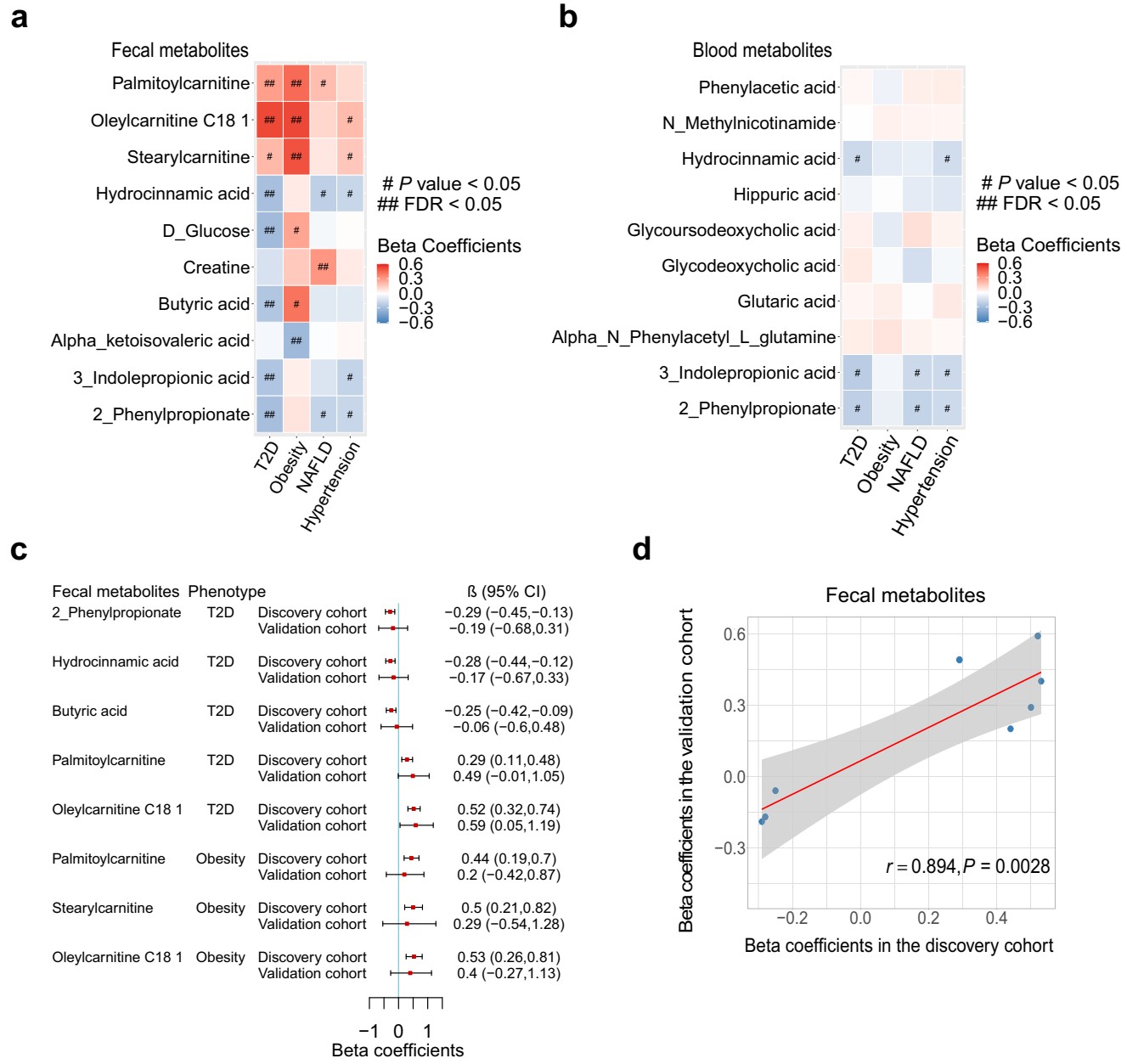

**Fig. 4 | The associations of well-predicted fecal and blood metabolites with cardiometabolic diseases. a** The associations of well-predicted fecal metabolites with cardiometabolic diseases. **b** The associations of well-predicted blood metabolites with cardiometabolic diseases. In **a**, **b**, the intensity of colors represents the partial regression coefficients that are computed by logistic models, adjusted by age, sex, smoking status, alcohol status, income, physical activity, and total energy intake for obesity, and by age, sex, BMI, smoking status, alcohol status, education, income, physical activity, and total energy intake for T2D, hypertension and NAFLD. FDR is controlled by the Benjamini–Hochberg method. #*P* < 0.05, ##FDR < 0.05. **c** Replication of the significant associations of well-predicted fecal

metabolites with cardiometabolic diseases in the validation cohort (*n* = 1007 in the discovery cohort; *n* = 103 in the validation cohort). Error bars are partial regression coefficients with 95% confidence intervals. **d** The scatter plot demonstrates the partial regression coefficients for the associations of well-predicted fecal metabolites with cardiometabolic diseases obtained in the discovery (*x* axis) and validation cohort (*y* axis). The correlation between the partial regression coefficients obtained in the discovery and validation cohort is computed by *Pearson* correlation. Error band is linear regression line with 95% confidence band. All statistical tests are two-sided. Source data are provided as a Source Data file. T2D, type 2 diabetes; NAFLD nonalcoholic fatty liver disease, FDR false discovery rate, CI confidence interval.

40–75 years living in the urban area of Guangzhou, China, for at least 5 years were recruited between 2008–2013. We followed these participants every three years. During 2014–2018 follow-up visits, stool and fasting blood samples were collected at the same time. We included participants with paired fecal and blood samples (*N* = 1008), and one paired fecal and blood samples were available per individual. We excluded participants who have taken antibiotics within 2 weeks (*N* = 1). Finally, 1007 participants (age: 64.7 ± 5.6; BMI: 23.6 ± 3.1) were remained for subsequent analysis (Table 1).

We used the control arm of a case-control study for hip fraction as the validation cohort[24]. The participants were enrolled between 2009 and 2012 in Guangdong Province, China. Stool and fasting blood samples were collected at the follow-up visits between February 2017 and May 2017 at the same time point. We included participants with paired fecal and blood samples in the present study (*N* = 103; age: 71.5 ± 7.1; BMI: 24.2 ± 3.7; Table 1).

All participants involved in this study provided written informed consent prior to sample collection. The Ethics Committee of the

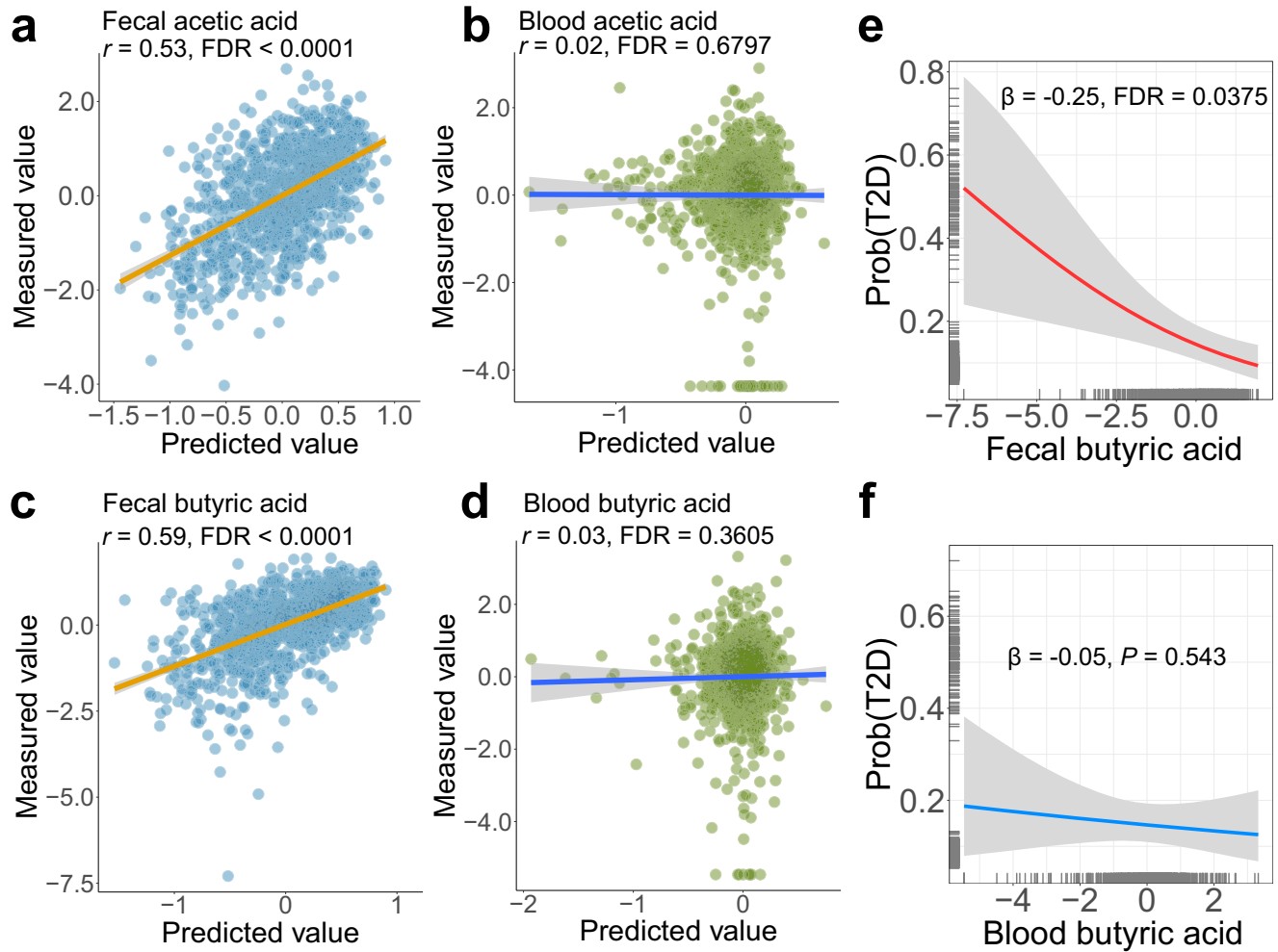

**Fig. 5 | Comparisons between paired fecal and blood SCFAs in associating taxonomic composition and type 2 diabetes. a** The association between taxonomic composition and fecal acetic acid. **b** The association between taxonomic composition and blood acetic acid. **c** The association between taxonomic composition and fecal butyric acid. **d** The association between taxonomic composition and blood butyric acid. The random forest model with five-fold cross-validation is used to predict the fecal or blood metabolite levels based on taxonomic composition. The scatter plot is plotted by the predicted and measured metabolite values. *Spearman*'s correlation between measured and predicted metabolite values is used to measure the association of taxonomic composition with fecal or blood metabolites. Error bands in **a–d** are linear regression lines with 95% confidence bands. **e** The association of fecal butyrate acid with T2D. **f** The association of blood butyrate acid with T2D. In **e**, **f**, logistic regression model is used to assess the associations of fecal or blood butyrate acid with T2D, adjusted for age, sex, BMI, smoking status, alcohol status, education, income, physical activity, and total energy intake. Error bands in **e**, **f** are logistic regression curves with 95% confidence bands. FDR is controlled by the Benjamini-Hochberg method. All statistical tests are two-sided. Source data are provided as a Source Data file. T2D, type 2 diabetes; SCFA, short-chain fatty acids; FDR, false discovery rate.

School of Public Health at Sun Yat-sen University (2018048) and Westlake University (20190114ZJS0003) approved the study protocols.

**Shotgun metagenomic sequencing and preprocessing**
Fecal samples from all participants in the discovery and validation cohort were collected on the examination day. During a follow-up visit to the study center, we gave each participant a stool sampler and provided the detailed instructions for stool sample collection. Participants collected their stool samples after defecation and delivered the sample to the staff immediately. The stool samples were temporarily stored in an ice box, and transported to the research laboratory and stored in a −80 °C freezer within four hours.

Microbial DNA was isolated using the QIAamp DNA Stool Mini Kit (Qiagen, Hilden, Germany) based on the manufacturer's instruction. DNA concentrations were determined by the Qubit quantification system (Thermo Scientific, Delaware, US). The Illumina HiSeq platform (Illumina Inc., CA, USA) was used for shotgun metagenome sequencing

with 2 × 300-bp paired-end reads protocol. The microbial DNA extraction and shotgun metagenome sequencing were performed at Novogene Company (Beijing, China). After sequencing, we obtained an average of 41.9 million (minimum: 22.1 million; maximum: 65.2 million) paired-end raw reads for each sample. The detailed information on bioinformatics analysis of the metagenome data could be found in our previous paper[37]. PRINSEQ (version 0.20.447) was employed to filter the reads with low-quality scores, with the following filtering parameters: (1) trim the reads by quality score from the 5' end and 3' end with a quality threshold of 20; (2) remove read pairs when either read was <60 bp, contained "N" bases or quality score mean bellow 30; and (3) deduplicate the reads. Reads that could be aligned to the human genome (H. sapiens, UCSC hg19) were removed (aligned with Bowtie2 v2.2.5 using –reorder –no-contain –dovetail)[38]. Taxonomic profiling of the metagenomic samples was performed using MetaPhlAn2 (version 2.6.02) with default parameters which used a library of clade-specific markers to provide pan-microbial (bacterial, archaeal, viral and eukaryotic) quantification at the species level[39]. We used the

## Table 1 | Characteristics of the study participants

| | GNHS cohort (N = 1007) | Validation cohort (N = 103) |
|---|---|---|
| Age, years, mean (SD) | 64.7 (5.6) | 71.5 (7.1) |
| Female, n (%) | 695 (69%) | 74 (71.8%) |
| BMI, kg/m², mean (SD) | 23.6 (3.1) | 24.2 (3.7) |
| Current smoker, n (%) | 75 (7.4) | 4 (3.9) |
| Current alcohol drinker, n (%) | 79 (7.8) | 9 (8.7) |
| **Education, n (%)** | | |
| Middle school or lower | 271 (26.9) | 10 (9.7) |
| High school or professional college | 452 (44.9) | 16 (15.5) |
| University | 284 (28.2) | 68 (66.0) |
| Unknown | 0 | 9 (8.7) |
| **Income level, n (%)** | | |
| Low (≤1500 ¥/month) | 253 (25.1) | 9 (8.7) |
| Middle (1501–3000 ¥/month) | 609 (60.5) | 24 (23.3) |
| High (>3000 ¥/month) | 145 (14.4) | 58 (56.3) |
| Unknown | 0 | 12 (11.7) |
| Physical activity, MET, mean (SD) | 41.2 (14.4) | 32.3 (5.8) |
| Total energy intake, kcal/day, mean (SD) | 1785.9 (547.3) | 1312.9 (361.4) |
| **Disease prevalence, n (%)** | | |
| T2D | 160 (15.9) | 27 (26.2) |
| Hypertension, n (%) | 472 (46.9) | 23 (22.3) |
| Obesity, n (%) | 86 (8.5) | 14 (13.6) |
| NAFLD, n (%) | 668 (66.3) | –[a] |
| **Medication use, n (%)** | | |
| T2D | 84 (8.3) | – |
| Hypertension | 301 (29.9) | – |
| Dyslipidemia | 258 (25.6) | – |

T2D type 2 diabetes, MET metabolic equivalent of task, NAFLD nonalcoholic fatty liver disease, SD standard deviation, BMI body mass index.
[a]Data were unavailable.

HUMAnN2 (version 2.8.1) with default parameters for functional profiling of metagenomic samples[40], in which microbial pathways were generated based on MetaCyc metabolic pathway database[41,42]. We included microbial species and pathways with a minimum detective relative abundance of 0.01% in at least 10% of the samples, which yielded 149 species and 214 pathways. We log-transformed the relative abundance of species and pathway features before subsequent analysis and scaled them to zero-mean and unit-variance.

### Fecal and blood metabolomics profiling and preprocessing

We performed targeted metabolomics profiling for fecal and serum samples by an ultra-performance liquid chromatography coupled to tandem mass spectrometry (UPLC-MS/MS) system (ACQUITY UPLC-Xevo TQ-S, Waters Corp., Milford, MA, 570 USA). The Q300 Kit provided by Metabo-Profile Corp. (Shanghai, China), coving up to 310 metabolites and 12 biochemical classes, was used for targeted metabolomics profiling, which was commonly used in several recent studies[43–45]. It mainly includes known gut microbiota-derived metabolites including SCFAs, bile acids, indoles, etc. and other key host metabolites, such as amino acids, carbohydrates, organic acids, and so on. Briefly, serum/lyophilized fecal sample vortexed vigorously with ice methanol (internal standards contained), and the supernatant was obtained. Then, ice-cold 50% methanol solution was added to dilute the sample with 4000 × g centrifugation, and the supernatant mixed with internal standards for each sample was sealed before UPLC-MS/

MS profiling. The instrument parameters were setting as follows: C18 analytical column (2.1 × 100 mm,1.7 µM); column temperature 40 °C; mobile phases A (water with 0.1% formic acid), mobile phases B (acetonitrile: IPA, 90:10). The whole profiling process was performed at Metabo-Profile Corp. (Shanghai, China). The quality control (QC) samples were made up of pooled samples and were run every 14 samples. Raw data generated by UPLC-MS/MS were processed using the QuanMET software (v2.0, Metabo-Profile, Shanghai, China) to perform peak integration, calibration, and quantification for each metabolite. Metabolomic features were annotated to metabolites with MSI level 1 of confidence by comparing them to the standards of targeted metabolites. The gut metagenomic sequencing and fecal metabolomics profiling were performed with independent randomization procedures to ensure that the sample orders were not the same.

We quantified the concentrations of 204 fecal metabolites and 232 blood metabolites in the targeted metabolomics measurements. There were 173 overlapped metabolites between fecal and blood metabolites. After removing fecal or blood metabolites that were detected in less than 80% of samples, 159 overlapped metabolites were obtained. We further excluded metabolites with the relative standard deviation (standard deviation/mean) value in QC samples larger than 0.3 in fecal or blood samples. Finally, 132 paired fecal and blood metabolites were included in the present study. These metabolites mainly include amino acids, fatty acids, organic acids, carbohydrates, bile acids, benzenoids, carnitines, phenylpropanoic acids, pyridines, indoles, organooxygen compounds, and nucleosides. We imputed the missing values of metabolites by half the minimal concentration of the corresponding metabolites in the remaining non-missing samples. We performed the log-transformation for fecal and blood metabolomics data and standardized them into Z-scores (mean = 0, variance = 1).

### Cardiometabolic disease ascertainment

Type 2 diabetes (T2D) was defined as fasting blood glucose ≥ 7.0 mmol/L (126 mg/dL) or HbA$_{1c}$ ≥ 6.5% (48 mmol/mol) or self-reported drug medications for T2D, according to T2D diagnosis criteria from the American Diabetes Association[46]. Hypertension was ascertained based on systolic blood pressure ≥ 140 mmHg or diastolic blood pressure ≥ 90 mmHg or current antihypertensive medication use, according to the hypertension diagnostic standards published by WHO/International Society of Hypertension Committee[47]. Nonalcoholic fatty liver disease (NAFLD) was identified based on abdominal ultrasonography using a Doppler sonography machine (Sonoscape SSI-5500, Shenzhen, China). NAFLD was diagnosed according to criteria of the Fatty Liver Disease Study Group of the Chinese Liver Disease Association[48]. Obesity was defined as BMI ≥ 28 based on the suggestion of Working Group On Obesity In China for Chinese populations[49].

### Statistical analysis

**Phenotypic correlation and genetic correlation analysis.** We estimated the phenotypic correlations (a direct comparison of each metabolite's values between feces and blood) between paired fecal and blood metabolites using partial *Spearman* correlation analysis, adjusted for age, sex and BMI. In addition, we calculated the genetic correlation (the proportion of shared heritability between paired fecal and blood metabolites) between fecal and blood metabolites using bivariate GREML analysis by GCTA[25,50]. Genetic correlation analysis was performed for 596 participants with matched genotyping and fecal and blood metabolomics data. The detailed information for the genotyping data was described in our previous paper[51]. We used the Benjamini-Hochberg method to control the false discovery rate (FDR) caused by multiple testing. Phenotypic or genetic correlations between paired fecal and blood metabolites with the absolute value of correlation coefficients |r|> 0.3 and FDR < 0.05 were considered significant.

**Gut microbiota-fecal/blood metabolome association analysis.** Given that there were strong symbiotic relationships among gut microbes, we treated the gut microbiota as a whole and estimated how well taxonomic composition or microbial pathways could predict the concentrations of fecal and blood metabolites using several machine learning pipelines. Random Forest (RF) model with default hyperparameters has recently been adopted to predict the fecal metabolites based on gut microbiota after comparing several machine learning pipelines[18]. Meanwhile, Light Gradient Boosting Machine (LightGBM) model has been used to estimate the associations of gut microbiota with serum metabolites[5]. As RF and LightGBM models were commonly used in the gut microbiota-metabolome association studies, we first used both of them to predict the levels of fecal and blood metabolites based on taxonomic composition or microbial pathways and compared their performances. The machine learning pipelines including the RF and LightGBM models were conducted using the five-fold cross-validation strategy to avoid the potential overfitting, with 202, 202, 201, 201, 201 held-out samples, respectively, for these five-folds. We calculated *Spearman*'s correlation coefficient $r$ and *Spearman*'s correlation $P$ value between the measured and predicted metabolite levels for held-out samples. *Spearman*'s correlation $P$ value was further transformed into FDR value to correct for multiple testing. We defined metabolites with $r > 0.3$ and FDR $< 0.05$ as well-predicted metabolites. The cut-off $r$ value was based on previous studies[18,26]. We also performed sensitivity analyses by setting the cut-off $r$ value as 0.2 and 0.4. Sensitivity analyses for participants without T2D medications ($N = 923$), hypertension medications ($N = 706$), dyslipidemia medications ($N = 749$), or without any of the above three medications ($N = 530$) were also performed, respectively. The RF and LightGBM methods were implemented using the R package randomForest (version: 4.6-14)[52] and lightgbm (version: 3.3.1)[53], respectively. We used the RF model with default hyperparameters recommended by Muller et al.'s study[18]. The hyperparameters of LightGBM model were determined according to Bar et al.'s study:[5] learning_rate = 0.005, max_depth = default, feature_fraction = 0.2, num_leaves = default, min_data_in_leaf = 15, metric = L2, early_stopping_rounds = None, n_estimators = 2000, bagging_fraction = 0.8, bagging_freq = 1. Eventually, as the RF model had a better performance in terms of root mean square error (RMSE) and predictability than LightGBM model (Supplementary Notes 1; Supplementary Fig. 1), the main results were reported based on the RF method throughout this study.

We assessed the differences between the associations of taxonomic composition/microbial pathways with paired fecal and blood metabolites for each well-predicted metabolite using the method proposed by Hittner et al.[54], which was implemented by R package cocor (version: 1.1-4)[55].

**Independent validation for the identified gut microbiota-fecal/blood metabolite associations.** We then attempted to replicate the above identified associations in an independent validation cohort. Firstly, the RF model was built in the GNHS (discovery) cohort to predict the levels of fecal/blood metabolites based on the taxonomic composition/microbial pathway data. Then, the constructed models were directly applied in the validation cohort, and the corresponding *Spearman*'s correlation coefficient $r$ and *Spearman*'s correlation FDR values between measured and predicted metabolite levels were obtained for each metabolite. We considered the associations with *Spearman*'s correlation coefficient $> 0.3$ and FDR $< 0.05$ as being validated.

**Associations between well-predicted fecal/blood metabolites and cardiometabolic diseases.** We examined the associations of well-predicted fecal and blood metabolites with prevalent cardiometabolic diseases (obesity [$N_{patients} = 86$], T2D [$N_{patients} = 160$], hypertension [$N_{patients} = 472$], and NAFLD [$N_{patients} = 668$]) using

multivariable logistic models, adjusted by age, sex, smoking status, alcohol status, education, income, physical activity, and total energy intake for obesity, and by age, sex, BMI, smoking status, alcohol status, education, income, physical activity, and total energy intake for T2D, hypertension and NAFLD. We also performed sensitivity analysis to additionally adjust for T2D, hypertension, and dyslipidemia medications in multivariable logistic models. Associations with FDR $< 0.05$ were considered statistically significant. The results were further replicated in the validation cohort. Only T2D ($N_{patients} = 27$), hypertension ($N_{patients} = 23$), and obesity ($N_{patients} = 14$) were available in the validation cohort. *Pearson* correlation between the partial regression coefficients obtained from the discovery and validation cohort was calculated.

All statistical analyses were performed using R software (version: 4.1.1) unless otherwise specified.

### Reporting summary
Further information on research design is available in the Nature Portfolio Reporting Summary linked to this article.

## Data availability
The raw data of metagenomic sequencing in this study have been deposited in the Genome Sequence Archive (GSA) (https://ngdc.cncb.ac.cn/gsa/) at accession number CRA008796. The fecal and serum metabolomics data have been deposited in the Metabolomics Workbench at study ID ST002337 and ST001669, respectively. UCSC hg19 is available from https://ftp.ebi.ac.uk/pub/databases/gencode/Gencode_human/release_19/GRCh37.p13.genome.fa.gz. The data associated with this study are presented in the paper (Supplementary Data 1-5 and Source Data). The metadata are available under restricted access due to participant consent and privacy regulations of our cohort, access can be obtained by request to the corresponding author (Yu-ming Chen: chenyum@mail.sysu.edu.cn) Source data are provided with this paper.

## Code availability
Analysis codes are available via https://github.com/nutrition-westlake/Paired-comparisons-between-the-fecal-and-blood-metabolites-in-their-associations-with-gut-microbiota/tree/main[56].

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

## Acknowledgements

We thank all study participants of the Guangzhou Nutrition and Health Study. We thank Westlake University Supercomputer Centre for assistance in data storage and computation. We thank the support from Westlake Education Foundation and Westlake Intelligent Biomarker Discovery Lab at the Westlake Laboratory of Life Sciences and Biomedicine. This study was funded by the National Natural Science Foundation of China (82073546 by Y.-m.C., 81903316 and 82073529 by J.-S.Z., 82103826 by Y.F., 82204161 by K.D.), China Postdoctoral Science Foundation (2022M710131 by K.D.), Zhejiang Provincial Natural Science Foundation of China (LQ21H260002 by Y.F.), Zhejiang Ten-thousand Talents Program (2019R52039 by J.-S.Z.) and the 5010 Program for Clinical Research (2007032 by Y.-m.C.) of the Sun Yat-sen University (Guangzhou, China).

## Author contributions

J.-S.Z., Y.-m.C. and K.D. contributed to study conceptualization and design. K.D., J.-j.X., L.S. and F.X. contributed to the data analysis. J.-j.X., B.-y.L. and W.H. contributed to data collection. H.Z., Y.F., M.S., and Z.J. contributed to data curation. K.D. and J.-S.Z. contributed to writing the manuscript. L.S., W.G., Y.F. and Z.J. contributed to the discussion. J.-S.Z., Y.-m.C., K.D., J.-j.X., and L.S. contributed to writing, reviewing, and editing the manuscript. All authors read, revised, and approved the final draft. J.-S.Z. and Y.-m.C. are the guarantors of this work and, as such, had full access to all the data in the study and take responsibility for the integrity of the data and the accuracy of the data analysis.

## Competing interests

The authors declare no competing interests.
