## [Peer Review File · Nature Communications]

REVIEWER COMMENTS

Reviewer #1 (Remarks to the Author):

Deng et al. present here an analysis of gut microbiome-associated metabolites, comparing metabolites measured in feces samples to their paired measurements in blood samples. They first compare metabolite levels between specimen and interestingly report mostly low correlations. They show that fecal metabolites are significantly more likely to associate with microbiome composition, compared with their respective blood measurements (as expected). They further show that certain microbial-associated metabolites in feces associate with one or more cardiometabolic conditions, but their respective blood measurements do not.

Overall, we found the paired feces-blood metabolite analysis interesting and potentially valuable for the microbiome research community, specifically in light of the relatively large cohort used here. The methodology is solid and straight forward. The manuscript in its current form, however, has several major caveats. Mainly, thorough language editing is required as many of the phrasings are vague, cumbersome, or grammatically incorrect. In addition, the take home message should be revised as it is currently misguided: The finding that fecal metabolites have stronger associations with both gut microbiota and cardiometabolic conditions does not imply that fecal metabolomics should be favored over blood metabolomics in clinical studies. Blood metabolites could still be key to understanding the mechanism underlying microbiota's involvement in promoting disease.

Major comments:

1. Extensive language editing is required. Many sentences in their current form are unclear, grammatically incorrect, or cumbersome. Specific examples (a non-exhaustive list):
 - a. The phrase “paired comparisons between the fecal and blood metabolites in their associations with...” is unclear. Lines 33-34, 134-135, etc.
 - b. In several places, replace “gut metagenomic, fecal and blood metabolomics data” to “gut metagenomic, and fecal and blood metabolomics data”. (lines 36,
 - c. Line 42: Change “findings could be replicated” to “findings were replicated” ...
 - d. Line 68-70: Inaccurate and unclear.
 - e. Line 105-106: Too vague. Be specific.

f. Line 126: likely meant “among most paired ...” (the word “most” is missing and crucial for sentence correctness).

g. Line 137-138: Should be rephrased to clarify that in this part of the analysis, the authors are independently examining the association between the gut microbiome and each metabolite, either in the feces sample or the blood sample. And that they then compare whether the association is consistent across feces/blood. Current analysis description is ambiguous.

h. Lines 185-187: it is not clear what “the above results were robust” mean. Please replace this phrase with what’s in the parenthesis.

i. Paragraph starting at 189 should be more concise. Specifically, it should be explained that what is shown is that not only that there were more well-predicted metabolites in the feces compared to blood, but also that models of well-predicted feces metabolites significantly outperformed their respective models of well-predicted blood metabolites.

j. Line 221: Seems redundant with previous paragraph. Rephrase.

k. Line 272: repetitive with previous paragraph.

2. The abstract is currently too vague and does not articulate well what the research question was.

a. The first sentence of the abstract (lines 32-33) is inaccurate and requires rephrasing. The blood metabolome is indeed shaped, in part, by the gut microbiome, but there are other, more influential factors (see reference 10 in your paper). Writing “widely used ... to represent the function of gut microbiota” is therefore an exaggeration.

b. Line 36-37: unclear what is meant by “low phenotypic and genetic correlations”. Reader should have a rough understanding of what analysis has been performed without reading into the Methods.

3. The second paragraph of the introduction mixes two different topics that should each be more clearly described. The first is what we know about feces-metabolome vs blood-metabolome associations to the gut microbiome. Here, for example, include estimations of variance explained by gut microbiome for each metabolome type (e.g. Tang, Zheng-Zheng, et al. *Frontiers in Genetics* (2019) or other references), and note the other factors shaping the metabolome in these specimens. The second topic is whether microbial associated metabolites promote cardiometabolic diseases (or protect from them) in the gut or through blood circulation. Specific examples should be provided.

4. When referring to methods or statistics that aren’t trivial, please either explain what you mean or refer the reader to the “Methods” section. Examples:

a. “relative standard deviation”, line 103 and 403

- b. Line 114: Explain that by ‘phenotypic correlations’ you mean a direct comparison of each metabolite’s values between fecal and blood.
- c. Line 125: Add a brief description of what is meant by “genetic correlations”.
- d. Line 144: Briefly describe how predictability is measured (as this is crucial for understanding subsequent statistics you provide).
- e. Lines 148-150: P value based on what test?
- f. Line 212: What is meant by “the above well-predicted metabolites”? Those validated in the independent cohort?
- g. Line 214: Associated using what test?

5. In the paragraph starting at line 200, clearly describe what is being validated. The current phrasing is ambiguous. By taking models trained on one cohort and applying them to another cohort, one can test whether these models are generalizable. It is not the same as testing whether specific metabolites are also well-predicted in another cohort (e.g. maybe they are well-predicted but using a completely different model?).

6. The discussion strongly advises to use fecal metabolome to better capture association with metabolic disease. However, capturing associations is often not the end-goal of these studies, but rather elucidating the mechanism by which the gut microbiome promotes diseases. The discussion should mention this point, and acknowledge the difference between association and causality (as is being hinted in line 310-312). The conclusion should be rephrased accordingly, as well. As mentioned above, the take home message is currently misguided: The finding that fecal metabolites have stronger associations with both gut microbiota and cardiometabolic conditions does not imply that fecal metabolomics should be favored over blood metabolomics in clinical studies. Blood metabolites could still be key to understanding the mechanism underlying microbiota’s involvement in promoting disease.

Minor comments:

1. In the paragraph starting at line 169 you explain that metabolites well-predicted by pathway features were a subset of those well-predicted by taxonomy features. This means that the taxonomy models outperformed pathway models in this case, and we recommend limiting the results presented from this point on only to the taxonomic models, to make this section less repetitive and more easy to read.

2. In the introduction, it needs to be clear that when “gut metagenomics” are discussed, this refers to fecal samples.
3. Line 54, phrase “human microbial community” should be replaced.
4. Metadata for each sample/individual should be published and referred to in the Data availability statement.
5. The term “gut microbiota” may be ambiguous when you specifically refer to the taxonomic features you used for machine learning modelling. It is better to use “gut microbiota composition” or “taxonomic composition”.
6. Inconsistent writing of numbers. For example, line 170 (“nine” instead of “9”), line 173, 174, etc.
7. Paragraph starting at 178 (discussing sensitivity analysis) should either be shortened or moved to the Methods, as the results are highly expected and do not add valuable information to the reader.
8. Line 255: It is inaccurate to write that blood metabolites are “frequently” used as microbial-related biomarkers. (Same inaccuracy as in the 1st sentence of the abstract).
9. Line 295: Associations between 2-phenylpropionate and T2D have been suggested (and not novel as currently claimed). One example: Gou, Wanglong, et al. Diabetes Care (2021).
10. Need to clarify in the Methods section whether more than one sample was analyzed per individual (e.g. did you include individuals with multiple time points for which blood+fecal samples were available?).
11. Line 433: Number of participants not consistent with what’s written in the “study cohorts” chapter. Explain why.
12. Few highly relevant recent papers that should be referenced in the introduction and discussed:

- a. Chen, Lianmin, et al. Nature Medicine (2022): 1-11.
- b. Dekkers, Koen F., et al. Nature communications 13.1 (2022): 1-12.
- c. Zeng, Xianfeng, et al. Cell 185.18 (2022): 3441-3456.

13. Consider increasing text size in some of the plots for readability.

Reviewer #2 (Remarks to the Author):

This study applies fecal metagenomics and targeted fecal and blood metabolomics in a discovery cohort of 1007 participants with replication in 103 participants. Sample size is larger than previous largest study (Zierer, 2018) however, the investigated number of metabolites much smaller. Findings supports earlier findings that there are stronger associations of microbiome with fecal metabolome than for plasma metabolome (Cox, Liver Int, 2022; Zhao, Journal of Translational Medicine 2022; Zierer, Nat Gen 2018). Researchers also assess the association of the two metabolomes with some prevalent disease (diabetes, NAFLD etc), however this part suffers from large limitations such as no adjustment for medication and the cross-sectional analysis. Moreover, methods do not describe procedures for assuring that fecal microbiota and metabolomics analysis order was not the same. If samples were analyzed in the same order/batches, that could heavily influence results. Conclusions are sometimes too strong, literature not well covered and the supplementary tables not in a good shape.

General

Grammar should be improved throughout the manuscript.

Introduction

The previous literature is not well covered. What did the previous study (15) show?

Methods

What was the rationale for the decision on excluding people with antibiotic treatment within only 2 weeks? Previous research has shown that effects can last much longer e.g. Abeles et al. (2016) showed persistent differences after 6 months in an interventional study.

Medication might affect the association of microbiota with metabolites (Weersma et al, Gut, 2020) .
Were prevalent medication taken into account in the analysis?

What was the rationale for correcting for BMI in the association between fecal and blood metabolites?

Provide details on how fecal samples were taken. Authors should adhere to the reporting recommendations such as the STORMS checklist. Were samples taken at home, at the clinic? Four hours is a lot of enzymatic activity. How does this affect bacterial growth and fecal metabolites? Perhaps this can make the fecal samples overly similar compared to the plasma samples.

Importantly, how were batch effects handled, was the analysis order scrambled for fecal microbiome compared to fecal metabolites analysis order - not to introduce a dependence there between the two datasets?

How were the metabolites selected for the targeted metabolites approach, no details are provided? It seems only common metabolites were included. Moreover, since selection for metabolites that are associated with the gut microbiome – can conclusions about metabolites in general be drawn?

How were metabolomics features annotated to metabolites? At what MSI level of confidence?

RF model was deemed to "had a better performance" - under what criteria?

How many hold-out samples?

"We calculated the correlations between the partial regression coefficients obtained in the discovery and validation cohort using the Pearson correlation analysis." Why Pearson here when Spearman everywhere else?

For the microbiome ~ metabolome associations validation means "Spearman's correlation (FDR < 0.05) and Cochran-Q heterogeneous test $P > 0.05$ ", but for the metabolite ~ cardiometabolic disease association it is "Cochran-Q test, $P > 0.05$ ". To use a heterogeneity test for validation is not good scientific practice. This test will not be significant if there are not huge differences. In my view, the

validation cohort does not replicate the findings of the discovery cohort and has huge CI's (Fig 4). This needs to be discussed. Which diseases were available in the validation cohorts and how many patients were there?

For cardiometabolic disease – clarify whether these are prevalent or incident. What about medication effect here - we know that medication has a major impact on the microbiota?

“Our study supported the beneficial role of butyric acid for T2D”. How can this be claimed in cross-sectional data, also without building a proper model investigating the role of medication. Causal language needs to be toned down.

Discussion

Literature not well covered in discussion.

Limitation section super short, does not cover lack of replication of cardiometabolic disease associations, lack of medication data in disease analysis etc. Saying that the targeted metabolomics focusing on microbiota-derived metabolites are representative of the metabolite spectrum needs better argumentation and foundation in data.

Figures/Tables

In my pdf, Fig 3 is low resolution, I cant read it.

Supplementary Tables, in general these are sloppy, contains several tables in each and are extremely difficult to understand.

Table S1 should be made into Table 1 and lacks information on the cardiometabolic disease prevalences, antibiotics and other medications.

GNHS and validation are rather different in education, income and diet, why is this? Is the energy intake of 1300 kcal in validation cohort plausible given the normal BMI?

ST2, P-values and FDR are provided as “0” for several traits, or missing. This is not informative.

ST3: I don't understand these tables – are the column names correct? I can find any species or pathways?

ST6: Provide OR if logistic regression was used.

Author's Response to Reviewer Comments

Reviewer reports:

Reviewer #1 (Remarks to the Author):

Deng et al. present here an analysis of gut microbiome-associated metabolites, comparing metabolites measured in feces samples to their paired measurements in blood samples. They first compare metabolite levels between specimen and interestingly report mostly low correlations. They show that fecal metabolites are significantly more likely to associate with microbiome composition, compared with their respective blood measurements (as expected). They further show that certain microbial-associated metabolites in feces associate with one or more cardiometabolic conditions, but their respective blood measurements do not.

Overall, we found the paired feces-blood metabolite analysis interesting and potentially valuable for the microbiome research community, specifically in light of the relatively large cohort used here. The methodology is solid and straight forward. The manuscript in its current form, however, has several major caveats. Mainly, thorough language editing is required as many of the phrasings are vague, cumbersome, or grammatically incorrect. In addition, the take home message should be revised as it is currently misguided: The finding that fecal metabolites have stronger associations with both gut microbiota and cardiometabolic conditions does not imply that fecal metabolomics should be favored over blood metabolomics in clinical studies. Blood metabolites could still be key to understanding the mechanism underlying microbiota's involvement in promoting disease.

Response: Thank you for your comments. We have performed thorough language editing in the revised manuscript (see details below). In addition, we agree that fecal metabolites have stronger associations with both gut microbiota and cardiometabolic conditions does not imply that fecal metabolomics should be favored over blood metabolomics in clinical studies, thus we have revised potentially misleading sentences in the revised manuscript (lines 353-356, lines 313-314).

Lines 353-356: "In epidemiological studies, fecal metabolites, compared with their blood counterpart, may be more relevant in providing a functional link between gut microbiota and cardiometabolic diseases."

Lines 313-314: "This further supported the strength of fecal metabolome in associating gut microbiota and metabolic diseases."

Major comments:

1. Extensive language editing is required. Many sentences in their current form are

unclear, grammatically incorrect, or cumbersome. Specific examples (a non-exhaustive list):

- a. The phrase “paired comparisons between the fecal and blood metabolites in their associations with...” is unclear. Lines 33-34, 134-135, etc.
- b. In several places, replace “gut metagenomic, fecal and blood metabolomics data” to “gut metagenomic, and fecal and blood metabolomics data”. (lines 36,
- c. Line 42: Change “findings could be replicated” to “findings were replicated”...
- d. Line 68-70: Inaccurate and unclear.
- e. Line 105-106: Too vague. Be specific.
- f. Line 126: likely meant “among most paired ...” (the word “most” is missing and crucial for sentence correctness).
- g. Line 137-138: Should be rephrased to clarify that in this part of the analysis, the authors are independently examining the association between the gut microbiome and each metabolite, either in the feces sample or the blood sample. And that they then compare whether the association is consistent across feces/blood. Current analysis description is ambiguous.
- h. Lines 185-187: it is not clear what “the above results were robust” mean. Please replace this phrase with what’s in the parenthesis.
- i. Paragraph starting at 189 should be more concise. Specifically, it should be explained that what is shown is that not only that there were more well-predicted metabolites in the feces compared to blood, but also that models of well-predicted feces metabolites significantly outperformed their respective models of well-predicted blood metabolites.
- j. Line 221: Seems redundant with previous paragraph. Rephrase.
- k. Line 272: repetitive with previous paragraph.

Response: Thank you for your very helpful language editing suggestions. We have revised the related sentences in the revised manuscript and revised supplementary materials.

For a:

In the revised manuscript:

Lines 35-36: “we performed comparisons between paired fecal and blood metabolites in their associations with gut microbiota and cardiometabolic diseases”

Lines 85-87: “Therefore, direct comparison between paired fecal and blood metabolome for their roles in linking the gut microbiota and cardiometabolic diseases has been rare but highly warranted”

Lines 148-149: “Comparison between paired fecal and blood metabolome in their associations with taxonomic composition and microbial pathways”

Lines 250-251: “Through comparing paired fecal and blood SCFAs in their associations with taxonomic composition”

Lines 263-264: “we conducted comparison between paired fecal and blood metabolome in their associations with gut microbiota and cardiometabolic diseases.”

Lines 279-281: “our study was the largest study comparing simultaneously collected paired fecal and blood metabolome for their associations with gut microbiota”

Lines 322-323: “which facilitates paired fecal and blood metabolome comparison.”

Lines 349-351: “In conclusion, we conduct comparison between paired fecal and blood metabolome in their associations with gut microbiota and cardiometabolic diseases in a large cohort study.”

Lines 860-861: “Fig. 3. Comparisons between paired fecal and blood metabolites in their associations with taxonomic composition”

Lines 908-909: “Fig. 5. Comparisons between paired fecal and blood SCFAs in associating taxonomic composition and type 2 diabetes”

In the revised Supplementary Figures:

Lines 51-52: “Fig. S4. Comparisons between paired fecal and blood metabolites in their associations with microbial pathways”

Lines 78-79: “Fig. S5. Comparisons between paired fecal and blood metabolites in their associations with taxonomic composition/microbial pathways”

Lines 149-150: “Fig. S10. Comparisons between the associations of microbial pathways with paired fecal and blood SCFAs.”

For b:

Lines 35-38: “we performed comparisons between paired fecal and blood metabolites in their associations with gut microbiota and cardiometabolic diseases among 1007 middle-aged and elderly adults with matched gut metagenomic, and fecal and blood metabolomics data.”

Lines 109-111: “We used the shotgun metagenomic sequencing and targeted quantitative fecal and serum metabolomics profiling to generate metagenome, and fecal and blood metabolome data, respectively, for both discovery and validation cohort.”

Lines 320-322: “this study has a large sample size with matched gut metagenomic, and fecal and blood metabolomics data collected simultaneously within each participant”

For c:

Lines 43-44: “Most of our findings were replicated in an independent validation cohort

involving 103 adults”

For d:

Lines 65-68: “While previous evidence demonstrated that gut microbiota significantly explained the variance of blood metabolome (average explained variance: 4.6%~15%)^{5, 6, 7, 8}, other studies showed that the gut microbial composition was largely reflected by fecal metabolome (average explained variance: 67.7%)⁹”

For e:

Lines 114-117: “These metabolites mainly consisted of known gut microbiota-derived metabolites including SCFAs, bile acids, indoles, etc. and other key host metabolites, such as amino acids, carbohydrates, organic acids, and so on”

Lines 424-427: “It mainly includes known gut microbiota-derived metabolites including SCFAs, bile acids, indoles, etc. and other key host metabolites, such as amino acids, carbohydrates, organic acids, and so on.”

For f:

Lines 139-141: “Consistent with the results of phenotypic correlations, there were low genetic correlations among most paired fecal and blood metabolites”

For g:

Lines 152-157: “We used two machine learning models, random forest (RF) and Light Gradient Boosting Machine (LightGBM) with five-fold cross-validation to independently examine the association between taxonomic composition/microbial pathways and each metabolite, either in the fecal or blood samples, and then compared whether the associations were consistent across feces and blood.”

For h:

Lines 191-193: “For sensitivity analysis with different cut-off r values (correlation coefficient: 0.2 and 0.4), the number of well-predicted fecal metabolites was substantially more than that of the blood metabolites (Supplementary Fig. S5CD).”

For i:

Lines 198-206: “Not only were there more well-predicted metabolites in the feces compared to the blood, but most well-predicted fecal metabolites had superior predictability over their corresponding paired blood metabolites (FDR < 0.05 by Cochran-Q test; Fig. 3D, Supplementary Fig. S4D and Fig. S7 and Supplementary Table S3). Moreover, the models of several well-predicted fecal metabolites significantly outperformed their respective models of well-predicted blood metabolites either based on taxonomic composition (FDR < 0.05 by Cochran-Q test; Fig. 3D and Supplementary Table S3) or microbial pathways (FDR < 0.05 by Cochran-Q test; Supplementary Fig. S4D and Supplementary Table S3).”

For j:

Lines 233-234: “These findings were interesting for well-predicted metabolites that were identified in both fecal and blood samples.”

For k:

We have deleted this repetitive sentence.

In addition to revising above sentences listed by the reviewer, we have also revised other sentences that have incorrect grammar or are unclear in the revised manuscript (lines 251-255, lines 262-263, lines 273-274, lines 293-294, lines 498-500, lines 557-558, lines 828-831, lines 850-852, lines 870-871, lines 919-920), and revised Supplementary Figures (line 2, lines 61-62, lines 102-103, lines 103-104, Lines 106-107, lines 136-138).

In the revised manuscript:

Lines 251-255: “we found that gut microbes were strongly associated with fecal levels of acetic acid ($r=0.53$, $FDR < 0.001$; Fig. 5A) and butyric acid ($r=0.59$, $FDR < 0.001$; Fig. 5C), but were not associated with blood acetic acid ($r=0.02$, $FDR=0.68$; Fig. 5B) or butyric acid ($r=0.03$, $FDR=0.36$; Fig. 5D).”

Lines 262-263: “In the present population-based large-scale gut microbiota-metabolome association study”

Lines 273-274: “The importance of fecal metabolome for understanding the gut microbiota function in humans has been recognized in the past few years”

Lines 293-294: “they may be maintained within physiological ranges by homeostasis that is achieved substantially through mass action-driven oxidation”

Lines 498-500: “we first used both of them to predict the levels of fecal and blood metabolites based on taxonomic composition or microbial pathways and compared their performances”

Lines 557-558: “Associations with Cochran-Q P value > 0.05 , $I^2 < 25\%$ and the same effect directions were considered as being validated”

Lines 828-831: “We estimate the gut microbiota-fecal/blood metabolome associations using the machine learning pipeline, and systematically compare the associations of taxonomic composition/microbial pathways with paired fecal and blood metabolites”

Lines 850-852: “there are 38 metabolites that are not converged during the calculation

process and thus had no results”

Lines 870-871: “Heterogeneity between the associations of taxonomic composition with paired fecal and blood metabolites”

Lines 919-920: “Logistic regression model is used to assess the associations of fecal or blood butyrate acid with T2D”

In the revised supplementary figures:

Line 2: “Fig. S1. Performance comparison of the RF and LightGBM model”

Lines 61-62: “Heterogeneity between the associations of microbial pathways with paired fecal and blood metabolites for well-predicted metabolites”

Lines 102-103: “Fig. S7. Heterogeneity between the associations of gut microbiota with paired fecal and blood metabolites”

Lines 103-104: “(A) Heterogeneity between the associations of taxonomic composition with paired fecal and blood metabolites”

Lines 106-107: “(B) Heterogeneity between the associations of microbial pathways with paired fecal and blood metabolites”

Lines 136-138: “The associations between taxonomic composition/microbial pathways and well-predicted fecal/blood metabolites are measured by *Spearman’s* correlation between measured and predicted metabolite levels obtained by RF model.”

2. The abstract is currently too vague and does not articulate well what the research question was.

a. The first sentence of the abstract (lines 32-33) is inaccurate and requires rephrasing. The blood metabolome is indeed shaped, in part, by the gut microbiome, but there are other, more influential factors (see reference 10 in your paper). Writing “widely used ... to represent the function of gut microbiota” is therefore an exaggeration.

b. Line 36-37: unclear what is meant by “low phenotypic and genetic correlations”. Reader should have a rough understanding of what analysis has been performed without reading into the Methods.

Response: To clearly articulate what the research question was, we have revised the related sentences (lines 33-38).

Lines 33-38: “To investigate and compare the roles of fecal and blood metabolites in linking the gut microbiota and cardiometabolic diseases, we performed comparisons between paired fecal and blood metabolites in their associations with gut microbiota

and cardiometabolic diseases among 1007 middle-aged and elderly adults with matched gut metagenomic, and fecal and blood metabolomics data.”

For a: We have rephrased the first sentence of the abstract.

Lines 32-33: “Blood metabolome has been used in clinical studies to explore the associations of gut microbiota-derived metabolites with cardiometabolic diseases”

For b: We have revised the related sentence.

Line 38-39: “Most paired fecal and blood metabolites had low correlations in the *Spearman* correlation analysis”

3. The second paragraph of the introduction mixes two different topics that should each be more clearly described. The first is what we know about feces-metabolome vs blood-metabolome associations to the gut microbiome. Here, for example, include estimations of variance explained by gut microbiome for each metabolome type (e.g. Tang, Zheng-Zheng, et al. *Frontiers in Genetics* (2019) or other references), and note the other factors shaping the metabolome in these specimens. The second topic is whether microbial associated metabolites promote cardiometabolic diseases (or protect from them) in the gut or through blood circulation. Specific examples should be provided.

Response: Thank you for your suggestions. The second paragraph of the introduction has been revised and separated into two paragraphs in the revised manuscript (lines 64-90).

Lines 64-90: “Feces and blood are both commonly used biological samples to capture the gut microbial metabolites. While previous evidence demonstrated that gut microbiota significantly explained the variance of blood metabolome (average explained variance: 4.6%~15%)^{5, 6, 7, 8}, other studies showed that the gut microbial composition was largely reflected by fecal metabolome (average explained variance: 67.7%)⁹. In addition, diet and genetics are also important factors shaping blood and fecal metabolome^{5, 6, 8, 9, 10, 11}. In epidemiological studies, blood was the most widely used biological sample for metabolome profiling to facilitate the exploration of the association between gut microbiota-related metabolites and cardiometabolic diseases^{12, 13, 14, 15, 16, 17}. For example, Nemet et al. found that blood phenylacetylglutamine was positively associated with the risk of cardiovascular disease and incident major adverse cardiovascular events¹⁴. Vangipurapu et al. identified several blood microbiota-related metabolites including kynurenate, dimethylglycine, 2-hydroxyhippurate that were associated with the risk of type 2 diabetes (T2D)¹². Circulating trimethylamine N-oxide was associated with cardiovascular disease risk¹⁷.

Currently, majority of existing gut microbiota-metabolome association studies were performed using either feces or blood for metabolome profiling^{5, 6, 9, 18, 19}. Although both fecal and blood metabolomics data were used in several prior studies to explore the gut microbiota-metabolome associations, fecal and blood samples were not

collected at the same time points and obtained from different subsets of participants²⁰, or the sample size was very small ($N < 100$)^{10,21}. Therefore, direct comparison between paired fecal and blood metabolome for their roles in linking the gut microbiota and cardiometabolic diseases has been rare but highly warranted, as it could help guide the practical applications of fecal and blood metabolomics technology in studying the function of gut microbiota for cardiometabolic diseases in human studies.”

4. When referring to methods or statistics that aren't trivial, please either explain what you mean or refer the reader to the “Methods” section. Examples:

- a. “relative standard deviation”, line 103 and 403
- b. Line 114: Explain that by ‘phenotypic correlations’ you mean a direct comparison of each metabolite’s values between fecal and blood.
- c. Line 125: Add a brief description of what is meant by “genetic correlations”.
- d. Line 144: Briefly describe how predictability is measured (as this is crucial for understanding subsequent statistics you provide).
- e. Lines 148-150: P value based on what test?
- f. Line 212: What is meant by “the above well-predicted metabolites”? Those validated in the independent cohort?
- g. Line 214: Associated using what test?

Response: Thank you for your suggestions. We have explained these methods or statistics in the revised manuscript.

For a:

Lines 111-113: “After excluding metabolites with the proportion of the missing value ≥ 0.2 or with relative standard deviation (standard deviation/mean) ≥ 0.3 in quality control samples”

Lines 448-449: “We further excluded metabolites with the relative standard deviation (standard deviation/mean) value in QC samples larger than 0.3 in fecal or blood samples.”

For b:

Lines 124-127: “We first assessed the phenotypic correlations (a direct comparison of each metabolite’s values between feces and blood) between paired fecal and blood metabolites using the partial Spearman correlation analysis adjusted for age, sex, and BMI.”

Lines 475-477: “We estimated the phenotypic correlations (a direct comparison of each metabolite’s values between feces and blood) between paired fecal and blood metabolites using partial *Spearman* correlation analysis”

For c:

Lines 137-139: “We then examined the genetic correlations (the proportion of shared

heritability between paired fecal and blood metabolites) between the paired fecal and blood metabolites using bivariate GREML analysis”

Lines 477-480: “In addition, we calculated the genetic correlation (the proportion of shared heritability between paired fecal and blood metabolites) between fecal and blood metabolites using the bivariate GREML analysis by GCTA”

For d:

Lines 160-164: “The predictability of fecal metabolites based on taxonomic composition or microbial pathways, which was measured by *Spearman’s* correlation between the measured and predicted metabolite levels for hold-out samples by machine learning pipeline, was much higher than those of blood metabolites”

For e:

Lines 165-169: “Overall, compared with blood metabolites, fecal metabolites had significantly higher correlation with taxonomic composition (correlation coefficients: mean±sd: 0.40±0.15 vs. 0.09±0.11; $P < 0.0001$ by Wilcoxon signed-rank test; Fig. 3C) and microbial pathways (correlation coefficients: mean±sd: 0.32±0.14 vs. 0.06±0.11; $P < 0.0001$ by Wilcoxon signed-rank test; Supplementary Fig. S4C).”

For f:

“the above well-predicted metabolites” refer to the identified well-predicted metabolites in the discovery cohort. We have clarified it in the revised manuscript (lines 221-223).

Lines 221-223: “Subsequently, we explored the associations of the above well-predicted fecal and blood metabolites identified in the discovery cohort with the prevalence of cardiometabolic diseases (T2D, obesity, NAFLD, and hypertension).”

For g:

Lines 223-227: “There were 12 well-predicted fecal metabolites being associated with several cardiometabolic diseases including T2D (7 significant associations), obesity (4 significant associations) and NAFLD (1 significant association) ($FDR < 0.05$ by the multivariable logistic models; Fig. 4A and Supplementary Table S5).”

We also described the methods used for calculating P values or refer the reader to the “Methods” section for other methods:

Lines 198-201: “Not only were there more well-predicted metabolites in the feces compared to the blood, but most well-predicted fecal metabolites had superior predictability over their corresponding paired blood metabolites ($FDR < 0.05$ by Cochran-Q test; Fig. 3D, Supplementary Fig. S4D and Fig. S7 and Supplementary Table S3).”

Lines 201-206: “the models of several well-predicted fecal metabolites significantly outperformed their respective models of well-predicted blood metabolites either based on taxonomic composition (FDR < 0.05 by Cochran-Q test; Fig. 3D and Supplementary Table S3) or microbial pathways (FDR < 0.05 by Cochran-Q test; Supplementary Fig. S4D and Supplementary Table S3).”

Lines 212-213: “Based on the Spearman’s correlation (FDR < 0.05) and Cochran-Q heterogeneous test $P > 0.05$ (Methods)”

5. In the paragraph starting at line 200, clearly describe what is being validated. The current phrasing is ambiguous. By taking models trained on one cohort and applying them to another cohort, one can test whether these models are generalizable. It is not the same as testing whether specific metabolites are also well-predicted in another cohort (e.g. maybe they are well-predicted but using a completely different model?).

Response: We are sorry that we did not make it clear. What is being validated is the association between taxonomic composition/microbial pathways and fecal/blood metabolites identified in the discovery cohort (that is, the models that predict the metabolite levels based on taxonomic composition/microbial pathways), which is equivalent to testing whether the models are generalizable in the validation cohort. The strategy we used for validation process is similar to Bar *et al.*’s study¹. To make it clear, we have revised the related sentence in the revised manuscript (lines 208-210, lines 213-217, lines 538-540, lines 879-882 and lines 884-886) and revised supplementary figures (lines 69-72, lines 75-76, lines 128-143).

In the revised manuscript:

Lines 208-210: “We then validated the robustness of the above associations between taxonomic composition/microbial pathways and well-predicted metabolites in an independent validation cohort.”

Lines 213-217: “79.9% (155/194) predictions, including 73.5% (72/98) for well-predicted fecal metabolites and 100% (10/10) for well-predicted blood metabolites based on taxonomic composition, and 83.1% (64/77) for well-predicted fecal metabolites and 100% (9/9) for well-predicted blood metabolites based on microbial pathways, could be validated”

Lines 538-540: “We considered the associations with *Spearman*’s correlation FDR < 0.05 and Cochran-Q test $P > 0.05$ as being validated.”

Lines 879-882: “(E) The number of well-predicted fecal and blood metabolites based on taxonomic composition and the number of validated associations between taxonomic composition and well-predicted fecal/blood metabolites in the validation cohort.”

Lines 884-886: “Associations with *Spearman*’s correlation $FDR < 0.05$ and Cochran-Q test $P > 0.05$ are considered as being validated.”

In the revised supplementary figures:

Lines 69-72: “(E) The number of well-predicted fecal and blood metabolites based on microbial pathways and the number of validated associations between microbial pathways and well-predicted fecal/blood metabolites in the validation cohort”

Lines 75-76: “Associations with *Spearman*’s correlation $FDR < 0.05$ and Cochran-Q test $P > 0.05$ are considered as being validated”

Lines 128-143: “Fig. S9. The associations between taxonomic composition/microbial pathways and well-predicted fecal/blood metabolites in the GNHS (discovery) and validation cohorts. (A) The associations between taxonomic composition and well-predicted fecal metabolites in the GNHS and validation cohort. (B) The associations between taxonomic composition and well-predicted blood metabolites in the GNHS and validation cohort (C) The associations between microbial pathways and well-predicted fecal metabolites in the GNHS and validation cohort. (D) The associations between microbial pathways and well-predicted blood metabolites in the GNHS and validation cohort. The associations between taxonomic composition/microbial pathways and well-predicted fecal/blood metabolites are measured by *Spearman*’s correlation between measured and predicted metabolite levels obtained by RF model. Cochran-Q test is used to test the heterogeneity between the associations obtained in the GNHS and validation cohort. Only validated taxonomic composition/microbial pathways-fecal/blood metabolite associations are presented. Associations with *Spearman*’s correlation $FDR < 0.05$ and Cochran-Q test $P > 0.05$ are considered as being validated in the validation cohort.”

6. The discussion strongly advises to use fecal metabolome to better capture association with metabolic disease. However, capturing associations is often not the end-goal of these studies, but rather elucidating the mechanism by which the gut microbiome promotes diseases. The discussion should mention this point, and acknowledge the difference between association and causality (as is being hinted in line 310-312). The conclusion should be rephrased accordingly, as well. As mentioned above, the take home message is currently misguided: The finding that fecal metabolites have stronger associations with both gut microbiota and cardiometabolic conditions does not imply that fecal metabolomics should be favored over blood metabolomics in clinical studies. Blood metabolites could still be key to understanding the mechanism underlying microbiota’s involvement in promoting disease.

Response: Thank you for these important comments. We agree that “capturing associations is often not the end-goal of these studies, but rather elucidating the mechanism by which the gut microbiome promotes diseases”. We have mentioned this point (lines 315-318) and acknowledge the difference between association and causality

in the discussion section (lines 327-330). We have revised related sentences that advised to use fecal metabolome rather than blood metabolome to capture the associations of gut microbiota-related metabolites with cardiometabolic diseases in clinical studies (lines 281-283, lines 313-314). We have also revised the related sentences in the conclusion (lines 353-356).

Lines 315-318: “However, capturing associations is just the first step in epidemiological studies aiming at providing the directions for downstream mechanistic investigations as how the gut microbiota-derived metabolites affect cardiometabolic diseases, in which blood metabolome may play a more important role.”

Lines 327-330: “There are also several limitations. First, associations identified in this study are not necessarily causal. Specifically, those identified well-predicted metabolites based on gut microbiota are not necessarily derived from gut microbiota, although there are high correlations.”

Lines 281-283: “which could potentially provide the guidance for the practical applications of fecal and blood metabolome in exploring the association of gut microbiota-related metabolites for the host.”

Lines 313-314: “This further supported the strength of fecal metabolome in associating gut microbiota and metabolic diseases.”

Lines 353-356: “In epidemiological studies, fecal metabolites, compared with their blood counterpart, may be more relevant in providing a functional link between gut microbiota and cardiometabolic diseases.”

Minor comments:

1. In the paragraph starting at line 169 you explain that metabolites well-predicted by pathway features were a subset of those well-predicted by taxonomy features. This means that the taxonomy models outperformed pathway models in this case, and we recommend limiting the results presented from this point on only to the taxonomic models, to make this section less repetitive and more easy to read.

Response: Thank you for your suggestions. We have revised the related paragraphs (lines 186-190).

Lines 186-190: “The identified well-predicted metabolites based on microbial pathways showed consistent results, in which all identified well-predicted metabolites based on microbial pathways belonged to the identified well-predicted metabolites based on taxonomic composition (Supplementary Fig. S4; Supplementary Table S2; Supplementary Fig. S5A and S5B).”

2. In the introduction, it needs to be clear that when “gut metagenomics” are discussed, this refers to fecal samples.

Response: We have revised the related sentence (lines 94-96).

Lines 94-96: “with multi-omics datasets of gut metagenomic sequencing (using fecal samples), targeted quantitative fecal and blood metabolome collected at the same time point.”

3. Line 54, phrase “human microbial community” should be replaced.

Response: “human microbial community” has been replaced by “Human gut microbiome” (line 54).

Line 54: “Human gut microbiome is a diverse and complex ecosystem”

4. Metadata for each sample/individual should be published and referred to in the Data availability statement.

Response: We understand this reviewer’s request about the data sharing. We have shared our metabolomics data and metagenomics data in the corresponding websites/database as you can see in the Data availability statement. However, participant consent and privacy regulations for Guangzhou Nutrition and Health Study prevented us directly sharing the metadata for each participant publicly. This situation is similar to many published papers in the Nature (PMID: 35418674) ² or Nature communications (PMID: 33020474, 31653850) ^{3, 4}. Nevertheless, metadata for each participant can be available from the corresponding authors (Yu-ming Chen: chenyum@mail.sysu.edu.cn) on reasonable request. We have specified it in the Data availability statement (lines 577-579).

Lines 577-579: “Other datasets used and/or analyzed during the current study (including the metadata) are available from the corresponding author on reasonable request.”

5. The term “gut microbiota” may be ambiguous when you specifically refer to the taxonomic features you used for machine learning modelling. It is better to use “gut microbiota composition” or “taxonomic composition”.

Response: We have replaced “gut microbiota” with “taxonomic composition” in the

related sentences and figures in the revised manuscript and revised supplementary figures.

6. Inconsistent writing of numbers. For example, line 170 (“nine” instead of “9”), line 173, 174, etc.

Response: We have revised the inconsistent numbers (lines 180-182, lines 184-186, lines 223-226).

Lines 180-182: “There were 10 well-predicted blood metabolites (Fig. 3B, Fig. 3E and Supplementary Table S2), among which, 8 were overlapped with fecal samples”

Lines 184-186: “There were 90 well-predicted metabolites that were identified only in the fecal samples and 2 (alpha-N-Phenylacetyl-L-glutamine and hippuric acid) only in the blood.”

Lines 223-226: “There were 12 well-predicted fecal metabolites being associated with several cardiometabolic diseases including T2D (7 significant associations), obesity (4 significant associations) and NAFLD (1 significant association)”

7. Paragraph starting at 178 (discussing sensitivity analysis) should either be shortened or moved to the Methods, as the results are highly expected and do not add valuable information to the reader.

Response: We have shortened the related paragraph (lines 191-193).

Lines 191-193: “For sensitivity analysis with different cut-off r values (correlation coefficient: 0.2 and 0.4), the number of well-predicted fecal metabolites was substantially more than that of the blood metabolites (Supplementary Fig. S5CD).”

8. Line 255: It is inaccurate to write that blood metabolites are “frequently” used as microbial-related biomarkers. (Same inaccuracy as in the 1st sentence of the abstract).

Response: We have revised the related sentences (lines 266-269, lines 32-33).

Lines 266-269: “We further showed that some blood metabolites, which were used to explore the role of microbial-derived metabolites for the host, were weakly associated with gut microbiota or cardiometabolic diseases.”

Lines 32-33: “Blood metabolome has been used in clinical studies to explore the associations of gut microbiota-derived metabolites with cardiometabolic diseases.”

9. Line 295: Associations between 2-phenylpropionate and T2D have been suggested (and not novel as currently claimed). One example: Gou, Wanglong, et al. Diabetes Care (2021).

Response: Thank you for your comments. We have deleted the related sentence that claimed it was novel.

10. Need to clarify in the Methods section whether more than one sample was analyzed per individual (e.g. did you include individuals with multiple time points for which blood+fecal samples were available?).

Response: Only one paired fecal and blood samples were available per individual. We have clarified this in the Methods section of the revised manuscript (lines 365-367).

Lines 365-367: “We included participants with paired fecal and blood samples ($N=1008$), and one paired fecal and blood samples were available per individual.”

11. Line 433: Number of participants not consistent with what’s written in the “study cohorts” chapter. Explain why.

Response: Sorry for the typos. Among 1007 participants with paired fecal and blood samples, 596 had matched genotyping and fecal and blood metabolomics data. We have corrected it in the revised manuscript (lines 480-481).

Lines 480-481: “Genetic correlation analysis was performed for 596 participants with matched genotyping and fecal and blood metabolomics data.”

12. Few highly relevant recent papers that should be referenced in the introduction and discussed:

- a. Chen, Lianmin, et al. Nature Medicine (2022): 1-11.
- b. Dekkers, Koen F., et al. Nature communications 13.1 (2022): 1-12.
- c. Zeng, Xianfeng, et al. Cell 185.18 (2022): 3441-3456.

Response: Done

13. Consider increasing text size in some of the plots for readability.

Response: Done

Reviewer #2 (Remarks to the Author):

This study applies fecal metagenomics and targeted fecal and blood metabolomics in a discovery cohort of 1007 participants with replication in 103 participants. Sample size is larger than previous largest study (Zierer, 2018) however, the investigated number of metabolites much smaller. Findings supports earlier findings that there are stronger associations of microbiome with fecal metabolome than for plasma metabolome (Cox, *Liver Int*, 2022; Zhao, *Journal of Translational Medicine* 2022; Zierer, *Nat Gen* 2018). Researchers also assess the association of the two metabolomes with some prevalent disease (diabetes, NAFLD etc), however this part suffers from large limitations such as no adjustment for medication and the cross-sectional analysis. Moreover, methods do not describe procedures for assuring that fecal microbiota and metabolomics analysis order was not the same. If samples were analyzed in the same order/batches, that could heavily influence results. Conclusions are sometimes too strong, literature not well covered and the supplementary tables not in a good shape.

General

Grammar should be improved throughout the manuscript.

Response: We have significantly improved the grammar in the revised manuscript.

Introduction

The previous literature is not well covered. What did the previous study (15) show?

Response:

(1) We have covered more previous relevant literatures in the revised manuscript including Chen L, *et al.*, *Nat Med*, 2022 ⁵; Dekkers KF, *et al.*, *Nat Commun*, 2022 ⁶; Tang ZZ, *et al.*, *Front Genet*, 2019 ⁷; Zeng X, *et al.*, *Cell*, 2022 ⁸; Zhao L, *et al.*, *J Transl Med*, 2022 ⁹; Diener, *et al.*, *Nat Metab*, 2022 ¹⁰.

In addition, we have revised the related sentence to discuss some previous literatures (lines 81-85).

Lines 81-85: “Although both fecal and blood metabolomics data were used in several prior studies to explore the gut microbiota-metabolome associations, fecal and blood samples were not collected at the same time points and obtained from different subsets of participants ²⁰, or the sample size was very small ($N < 100$) ^{10, 21}”

(2) Previous study (Original literature number: 15; Current literature number: 20) showed that microbial pathways were associated with more fecal and blood metabolites compared with microbial species.

Methods

What was the rationale for the decision on excluding people with antibiotic treatment

within only 2 weeks? Previous research has shown that effects can last much longer e.g. Abeles et al. (2016) showed persistent differences after 6 months in an interventional study.

Response: Thank you for your comments. We agree that the effects of antibiotic treatments can last long time (6 months or even more longer). However, our cohort only surveyed the antibiotics use within 2 weeks at the time of stool sample collection to minimize the recall bias, as it is easier to recall the antibiotics use within a short time (e.g. 2 weeks). Thus, the findings in our study should be further validated in other large cohorts that surveyed the long-term antibiotics use (> 6 months). We have added this limitation in the revised manuscript (lines 333-337).

Lines 333-337: “Third, our cohort only surveyed the antibiotics use within 2 weeks at the time of stool sample collection. As the effects of antibiotic treatments can last longer time, our findings should be further validated in other large cohorts that surveyed the long-term antibiotics use (> 6 months).”

Medication might affect the association of microbiota with metabolites (Weersma et al, Gut, 2020) . Were prevalent medication taken into account in the analysis?

Response: Thank you for your comments. We now performed the sensitivity analysis for gut microbiota-metabolite associations among participants without T2D medications, hypertension medications, dyslipidemia medications, or without any of the above three medications, respectively, and obtained consistent results ($r = 0.99$, $P < 0.0001$ for participants without T2D medications; $r = 0.985$, $P < 0.0001$ for participants without hypertension medications; $r = 0.985$, $P < 0.0001$ for participants without dyslipidemia medications; $r = 0.976$, $P < 0.0001$ for participants without any of the above three medications). We have added these results in the revised supplementary figures (Supplementary Fig. S6), and added corresponding descriptions in the results and methods sections of the revised manuscript (lines 193-196, lines 509-512).

Lines 193-196: “Sensitivity analysis for gut microbiota-fecal/blood metabolite associations among participants without T2D, hypertension or dyslipidemia medications showed consistent results (Supplementary Fig. S6).”

Lines 509-512: “Sensitivity analyses for participants without T2D medications ($N = 923$), hypertension medications ($N = 706$), dyslipidemia medications ($N = 749$), or without any of the above three medications ($N = 530$) were also performed, respectively”

What was the rationale for correcting for BMI in the association between fecal and blood metabolites?

Response: We corrected for BMI in the associations between fecal and blood metabolites because several studies showed that BMI was associated with both fecal metabolites^{11, 12} and blood metabolites^{13, 14}, which may confound the association between paired fecal and blood metabolites. We have now conducted additional analysis without correcting for BMI in the associations between paired fecal and blood metabolites, and obtained consistent results ($r = 0.999$, $P < 0.0001$). We have added this result in the revised supplementary figures (Supplementary Fig. S3), and added corresponding descriptions in the revised manuscript (lines 134-135).

Lines 134-135: “Partial *Spearman* correlation analysis only adjusted for age and sex showed consistent results ($r = 0.999$, $P < 0.0001$; Supplementary Fig. S3).”

Provide details on how was fecal samples were taken. Authors should adhere to the reporting recommendations such as the STORMS checklist. Were samples taken at home, at the clinic? Four hours is a lot of enzymatic activity. How does this affect bacterial growth and fecal metabolites? Perhaps this can make the fecal samples overly similar compared to the plasma samples.

Response: Thank you for your suggestions. We have detailed the information on how fecal samples were taken in the revised manuscript (lines 385-390). The samples were taken at our study center. The stool samples were stored in an ice box during transportation, which could minimize the enzymatic activity and its influence on bacterial growth and fecal metabolites.

Lines 385-390: “During a follow-up visit to the study center, we gave each participant a stool sampler and provided the detailed instructions for stool sample collection. Participants collected their stool samples after defecation and delivered the sample to the staff immediately. The stool samples were temporarily stored in an ice box, and transported to the research laboratory and stored in a -80°C freezer within four hours.”

Importantly, how were batch effects handled, was the analysis order scrambled for fecal microbiome compared to fecal metabolites analysis order - not to introduce a dependence there between the two datasets?

Response: Thank you for your comments. The gut metagenomic sequencing and fecal metabolomics profiling were performed with independent randomization procedures to ensure that the sample orders were not the same. Therefore, our study would not introduce dependence between these two datasets. We have clarified it in the revised manuscript (lines 440-442).

Lines 440-442: “The gut metagenomic sequencing and fecal metabolomics profiling were performed with independent randomization procedures to ensure that the sample

orders were not the same.”

How were the metabolites selected for the targeted metabolites approach, no details are provided? It seems only common metabolites were included. Moreover, since selection for metabolites that are associated with the gut microbiome – can conclusions about metabolites in general be drawn?

Response:

(1) The targeted metabolomics analysis was performed using the Q300 Kit provided by Metabo-Profile Corp. (Shanghai, China), which was also used in several recent studies^{15, 16, 17}. The Q300 Kit covers up to 310 metabolites and 12 biochemical classes. It includes known gut microbiota-derived metabolites including SCFAs, bile acids, indoles, etc. and other key host metabolites, such as amino acids, carbohydrates, organic acids, and so on. We have added this information in the revised manuscript (lines 422-427).

Lines 422-427: “The Q300 Kit provided by Metabo-Profile Corp. (Shanghai, China), covering up to 310 metabolites and 12 biochemical classes, was used for targeted metabolomics profiling, which was commonly used in several recent studies^{43, 44, 45}. It mainly includes known gut microbiota-derived metabolites including SCFAs, bile acids, indoles, etc. and other key host metabolites, such as amino acids, carbohydrates, organic acids, and so on.”

(2) Metabolites contained in Q300 Kit covered a wide range of metabolic classes, including amino acids, fatty acids, organic acids, carbohydrates, bile acids, benzenoids, carnitines, phenylpropanoic acids, pyridines, indoles, organooxygen compounds, and nucleosides. Known gut microbiota-derived metabolites are only one part of them. Thus, our conclusions could be generalized to more broader metabolites to some extent. However, given that metabolites in Q300 kit constitute only a small fraction of the metabolic spectrum, our results should be further validated in large cohorts with more measured metabolites in the future. We have added this limitation in the revised manuscript (lines 332-333).

Lines 332-333: “Therefore, our results should be validated in large cohorts with a large number of measured metabolites in the future”

How were metabolomics features annotated to metabolites? At what MSI level of confidence?

Response: Metabolomic features were annotated to metabolites with MSI level 1 of confidence by comparing them to the standards of targeted metabolites. We have added this information in the revised manuscript (lines 439-440).

Lines 439-440: “Metabolomic features were annotated to metabolites with MSI level 1 of confidence by comparing them to the standards of targeted metabolites.”

RF model was deemed to "had a better performance" - under what criteria?

Response: RF model was deemed to "had a better performance" under the criteria of root mean square error and the number of identified well-predicted metabolites. Our results showed that RF model had a smaller root mean square error and could identify more well-predicted metabolites than LightGBM model. Thus, RF model had a better performance than the LightGBM model. We have clarified it in the revised manuscript (lines 157-159).

Lines 157-159: “As the RF model had a better performance than the LightGBM model with a smaller root mean square error and more well-predicted metabolites (Supplementary Fig. S1, Supplementary Note 1)”

How many hold-out samples?

Response: In the five-fold cross-validation of the machine learning model, the hold-out samples are 202, 202, 201, 201, 201, respectively, for these five-folds. We have added this information in the revised manuscript (lines 500-503).

Lines 500-503: “The machine learning pipelines including the RF and LightGBM models were conducted using the five-fold cross-validation strategy to avoid the potential overfitting, with 202, 202, 201, 201, 201 hold-out samples, respectively, for these five-folds.”

“We calculated the correlations between the partial regression coefficients obtained in the discovery and validation cohort using the Pearson correlation analysis.” Why Pearson here when Spearman everywhere else?

Response: Thank you for your comments. We used the *Pearson* correlation analysis to assess the consistency of the results, similar to previous studies^{10, 18, 19} (e.g. partial regression coefficients obtained from the discovery and validation cohorts, predictability results obtained by the RF and LightGBM models, gut microbiota-fecal/blood metabolite associations for all participants and those without medications, phenotypic correlations with and without adjusting for BMI, metabolites-cardiometabolic disease associations with and without adjusting for medications in this study). As *Pearson* correlation analysis directly measures the similarity of raw values rather than the similarity of ranks (transformed values from the raw values) as in the *Spearman* correlation analysis, it has a stronger power than *Spearman* correlation analysis to assess the consistency of the results, especially when the sample size is small

(e.g. only 8 pairs of partial regression coefficients obtained from the discovery and validation cohorts in this study). However, for paired fecal-blood metabolite correlations and measured-predicted metabolite correlations, *Spearman* correlation analysis is more suitable as it is robust to the distribution of the data.

In addition, to address this reviewer's concern, we now calculated the *Spearman's* correlations for partial regression coefficients obtained from the discovery and validation cohorts ($r = 0.810$, $P = 0.022$), predictability results obtained by the RF and LightGBM models ($r = 0.986$, $P < 0.0001$), gut microbiota-fecal/blood metabolite associations for all participants and those without medications ($r = 0.976$, $P < 0.0001$ for T2D medications; $r = 0.964$, $P < 0.0001$ for hypertension medications; $r = 0.966$, $P < 0.0001$ for dyslipidemia medications; $r = 0.947$, $P < 0.0001$ for all above three medications), phenotypic correlations with and without adjusting for BMI ($r = 0.990$, $P < 0.0001$), and metabolites-cardiometabolic disease associations with and without adjusting for medications ($r = 0.993$, $P < 0.0001$), and obtained similar significant results. We did not add the above results into the current version of the manuscript, but we can discuss how to integrate them into the manuscript if this reviewer has a strong preference to add these contents of *Spearman* correlation analysis.

For the microbiome ~ metabolome associations validation means "Spearman's correlation (FDR < 0.05) and Cochran-Q heterogeneous test $P > 0.05$ ", but for the metabolite ~ cardiometabolic disease association it is "Cochran-Q test, $P > 0.05$ ". To use a heterogeneity test for validation is not good scientific practice. This test will not be significant if there are not huge differences. In my view, the validation cohort does not replicate the findings of the discovery cohort and has huge CI's (Fig 4). This needs to be discussed. Which diseases were available in the validation cohorts and how many patients were there?

Response:

(1) Only the association between oleylcarnitine and T2D was fully replicated in the validation cohort. The low fully replication rate is largely a power issue, as the sample size of the validation cohort is small ($N_{total} = 103$; $N_{T2D} = 27$; $N_{hypertension} = 23$; $N_{Obesity} = 14$), which caused a large CI in Fig 4C. Therefore, instead of evaluating the fully replication results using P value (or CI), we assessed whether the effect sizes were homogeneous between the discovery and validation cohort, and this strategy was also used in previous studies with small validation cohorts^{19, 20}. To make the results of homogeneous assessment more robust, we now additionally calculated I^2 statistic to quantify the heterogeneity of the partial regression coefficients obtained from the discovery and validation cohort, and $I^2 < 25\%$ represents homogeneous results. Fig. 4C showed that I^2 for all associations are 0, which supports the homogeneous results obtained from the discovery and validation cohorts. Given that P for heterogeneity > 0.05 , $I^2 = 0$, and same directions of partial regression coefficients, our results could be considered as being validated, although not fully validated in the validation cohort. The

small sample size of the validation cohort is a limitation of this study, which lacks the sufficient statistical power to detect significant associations. We have discussed it in the revised manuscript (lines 337-342). We have also added the descriptions of I^2 statistic in the method and result sections of the revised manuscript (lines 239-244, lines 555-558, lines 898-901).

Lines 337-342: “Fourth, there are only 103 participants in the validation cohort with the number of patients 27, 23, and 14 for T2D, hypertension, and obesity, respectively, which limits the statistical power to fully replicate the results of metabolite-cardiometabolic disease associations. In addition, as NAFLD status is not available in the validation cohort, the corresponding associations are unable to validate. Thus, larger replication cohorts are needed in future to further replicate our findings.”

Lines 239-244: “We discovered that 8 out of 11 associations for the well-predicted fecal metabolites showed no significant differences in their partial regression coefficients between the discovery and validation cohorts (Cochran-Q test, $P > 0.05$; $I^2 = 0$; the same effect directions; $r = 0.894$; Fig. 4CD), which suggested that majority of our identified associations between well-predicted fecal metabolites and cardiometabolic diseases were robust.”

Lines 555-558: “We assessed the heterogeneity between the partial regression coefficients obtained in the discovery and validation cohort using the Cochran-Q test and I^2 statistic⁵⁶. Associations with Cochran-Q P value > 0.05 , $I^2 < 25\%$ and the same effect directions were considered as being validated.”

Lines 898-901: “The Cochran-Q test and I^2 statistic are used to assess the heterogeneity between the partial regression coefficients obtained in the discovery and validation cohort. Associations with Cochran-Q P value > 0.05 , $I^2 < 25\%$ and the same effect directions are considered as being validated.”

(2) Only type 2 diabetes, hypertension and obesity are available in the validation cohort, with the number of patients 27, 23, and 14, respectively. We have added the numbers of the patients in the discovery and validation cohorts in the revised manuscript (lines 544-550, lines 554-555).

Lines: 544-550: “We examined the associations of well-predicted fecal and blood metabolites with the prevalence of cardiometabolic diseases (obesity [$N_{\text{patients}} = 86$], T2D [$N_{\text{patients}} = 160$], hypertension [$N_{\text{patients}} = 472$], and NAFLD [$N_{\text{patients}} = 668$]) using multivariable logistic models, adjusted by age, sex, smoking status, alcohol status, education, income, physical activity, and total energy intake for obesity, and by age, sex, BMI, smoking status, alcohol status, education, income, physical activity, and total energy intake for T2D, hypertension and NAFLD.”

Lines: 554-555: “Only T2D ($N_{patients} = 27$), hypertension ($N_{patients} = 23$), and obesity ($N_{patients} = 14$) were available in the validation cohort”

For cardiometabolic disease – clarify whether these are prevalent or incident. What about medication effect here - we know that medication has a major impact on the microbiota?

Response:

(1) The prevalence of cardiometabolic diseases were explored in this study. We have now clarified it in the revised manuscript (lines 221-223, lines 544-545, lines 256-258).

Lines 221-223: “Subsequently, we explored the associations of the above well-predicted fecal and blood metabolites identified in the discovery cohort with the prevalence of cardiometabolic diseases (T2D, obesity, NAFLD, and hypertension).”

Lines 544-545: “We examined the associations of well-predicted fecal and blood metabolites with the prevalence of cardiometabolic diseases”

Lines 256-258: “Furthermore, we found that fecal butyric acid levels were inversely associated with the prevalence of T2D”

(2) We now performed the sensitivity analysis to additionally adjust for medications (T2D, hypertension and dyslipidemia medications) in the multivariable logistic models, and obtained the consistent results ($r = 0.998$, $P < 0.0001$). We have added the corresponding Fig. S8 in the revised supplementary figures, and added related sentences in the revised manuscript (lines 227-229, lines 550-552).

Lines 227-229: “Sensitivity analysis with an additional adjustment of medications for T2D, hypertension, and dyslipidemia showed consistent results ($r = 0.998$, $P < 0.0001$; Supplementary Fig. S8).”

Lines 550-552: “We also performed sensitivity analysis to additionally adjust for T2D, hypertension, and dyslipidemia medications in multivariable logistic models”

“Our study supported the beneficial role of butyric acid for T2D”. How can this be claimed in cross-sectional data, also without building a proper model investigating the role of medication. Causal language needs to be toned down.

Response: Thank you for your comments. We have deleted this sentence in the revised manuscript to avoid any confusion caused.

Discussion

Literature not well covered in discussion.

Response: We have now covered more relevant literatures in discussion section of the revised manuscript, for example, Chen L, et al., Nat Med, 2022 ⁵; Dekkers KF, et al., Nat Commun, 2022 ⁶; Diener, et al., Nat Metab, 2022 ¹⁰; Tang ZZ, et al., Front Genet, 2019 ⁷; Zhao L, et al., J Transl Med, 2022 ⁹. Jiaqian Qi, Journal of cellular and molecular medicine, 2018 ²¹.

In addition, we have added related sentences to discuss some previous literatures (lines 275-279).

Lines 275-279: “Although paired fecal and blood metabolome data are collected in some previous studies ^{10,21}, their samples sizes were very small ($N < 100$) and direct comparisons between paired fecal and blood metabolites in their associations with gut microbiota were not performed.”

Limitation section super short, does not cover lack of replication of cardiometabolic disease associations, lack of medication data in disease analysis etc. Saying that the targeted metabolomics focusing on microbiota-derived metabolites are representative of the metabolite spectrum needs better argumentation and foundation in data.

Response: We have now added more discussions in the limitation section of the revised manuscript based on the comments from reviewers (lines 327-346). In addition, we have deleted the sentences saying that our measured metabolites are representative of the metabolite spectrum in the revised manuscript.

Lines 327-346: “There are also several limitations. First, associations identified in this study are not necessarily causal. Specifically, those identified well-predicted metabolites based on gut microbiota are not necessarily derived from gut microbiota, although there are high correlations. Second, there are only 132 metabolites in our study because of the targeted metabolome profiling method. These metabolites constitute only a small fraction of metabolic spectrum. Therefore, our results should be validated in large cohorts with a large number of measured metabolites in the future. Third, our cohort only surveyed the antibiotics use within 2 weeks at the time of stool sample collection. As the effects of antibiotic treatments can last longer time, our findings should be further validated in other large cohorts that surveyed the long-term antibiotics use (> 6 months). Fourth, there are only 103 participants in the validation cohort with the number of patients 27, 23, and 14 for T2D, hypertension, and obesity, respectively, which limits the statistical power to fully replicate the results of metabolite-cardiometabolic disease associations. In addition, as NAFLD status is not available in the validation cohort, the corresponding associations are unable to validate. Thus, larger replication cohorts are needed in future to further replicate our findings. Finally, as the participants enrolled in our study are not selected at random, there is potential selection

bias in our study. Additionally, as our cohorts are only based on individuals of Chinese ethnicity, the generalization of our conclusions to populations with other ethnicities needs to be further investigated.”

Figures/Tables

In my pdf, Fig 3 is low resolution, I cant read it.

Response: We have improved the quality of this Figure, as well as other figures.

Supplementary Tables, in general these are sloppy, contains several tables in each and are extremely difficult to understand.

Response: Sorry for this. We have now put each Supplementary table in each separate excel to make them easy to read and understand.

Table S1 should be made into Table 1 and lacks information on the cardiometabolic disease prevalences, antibiotics and other medications.

Response: Thank you for your advice. We have now made Table S1 into Table 1, and added the information on cardiometabolic disease prevalences and medications in Table 1. As we have excluded the participants who have taken antibiotics within 2 weeks ($N=1$) before formal analysis, the information on antibiotics was not shown in Table 1.

GNHS and validation are rather different in education, income and diet, why is this? Is the energy intake of 1300 kcal in validation cohort plausible given the normal BMI?

Response:

(1) The distributions of education, income, and diet in GNHS and the validation cohort are different, because the participants from these two cohorts were recruited from different sources. Participants in GNHS are from general populations in the community. Participants in the validation cohorts are from the control arm of a case-control study for hip fracture, which were selected from apparently healthy community residents and inpatients who had been hospitalized within 1 week with one of the following diseases: influenza, pneumonia, benign ophthalmic or otorhinolaryngologic tumor, and acute surgical disease or cataract in one eye.

(2) The mean total energy intake of 1312.9 kcal in the validation cohort is plausible, given that participants in our study are older (mean age: 71.5) with most females (71.8%). Several studies based on elderly Chinese population also showed similar mean total energy intake value to our study. For example, one study with younger age (mean age: 69.5) and larger BMI (mean BMI: 25.9) than our study showed that the mean total energy intake for females was 1385 kcal/day²². Another study showed that mean total

energy intake for elderly participants (mean age: 70) was 1348 kcal/day²³.

ST2, P-values and FDR are provided as “0” for several traits, or missing. This is not informative.

Response: We have now rounded all values to four digits and replaced values less than 0.0001 as “< 0.0001”. Missing values in the results of genetic correlations occurred because there were 38 metabolites that did not converge during the calculation process and thus had no results. We have filled in missing blanks with “-” and explained it at the footnote (Supplementary Table S1).

ST3: I don’t understand these tables – are the column names correct? I can find any species or pathways?

Response: We are sorry that we did not make it clear. The results of this table are the predictability of fecal and blood metabolites based on taxonomic composition or microbial pathways, which was measured by *Spearman*’s correlation between the measured and predicted metabolite levels for hold-out samples by machine learning pipeline. To make it clear, we have changed the title of this table as “The predictability of fecal and blood metabolites based on taxonomic composition or microbial pathways using machine learning pipeline”, and changed the corresponding subtitles, and added the explanations at the footnotes (Supplementary Table S2).

ST6: Provide OR if logistic regression was used.

Response: Done (Supplementary Table S5).

Editorial requests:

as requested by the reviewers, please provide a detailed filled-in STORMS checklist as a Supplementary file along with the revisions, see here: <https://www.stormsmicrobiome.org/>. Nature Communications formally supports the inclusion of the STORMS checklist.

Response: We have filled in STORMS checklist for microbiome. To comply with this STORMS checklist, we have added related sentences in the revised manuscript (lines 398-400, lines 342-346, lines 396-398).

Lines 398-400: “After sequencing, we obtained an average of 41.9 million (minimum: 22.1 million; maximum: 65.2 million) paired-end raw reads for each sample.”

Lines 342-346: “Finally, as the participants enrolled in our study are not selected at random, there is potential selection bias in our study. Additionally, as our cohorts are only based on individuals of Chinese ethnicity, the generalization of our conclusions to populations with other ethnicities needs to be further investigated.”

Lines 396-398: “The microbial DNA extraction and shotgun metagenome sequencing were performed at Novogene Company (Beijing, China).”

Please complete or update the following checklist(s) to verify compliance with our research ethics and data reporting standards. Address all points on the checklist, revising your manuscript in response to the points if needed.

The form(s) must be downloaded and completed in Adobe Reader rather than opened in a web browser. Each form must be uploaded as a Related Manuscript file at the time of resubmission.

Response: Done.

Reference

1. Bar N, *et al.* A reference map of potential determinants for the human serum metabolome. *Nature* **588**, 135-140 (2020).
2. Gacesa R, *et al.* Environmental factors shaping the gut microbiome in a Dutch population. *Nature* **604**, 732-739 (2022).
3. Lee G, *et al.* Distinct signatures of gut microbiome and metabolites associated with significant fibrosis in non-obese NAFLD. *Nat Commun* **11**, 4982 (2020).
4. Boer CG, *et al.* Intestinal microbiome composition and its relation to joint pain and inflammation. *Nat Commun* **10**, 4881 (2019).
5. Chen L, *et al.* Influence of the microbiome, diet and genetics on inter-individual variation in the human plasma metabolome. *Nat Med*, (2022).
6. Dekkers KF, *et al.* An online atlas of human plasma metabolite signatures of gut microbiome composition. *Nat Commun* **13**, 5370 (2022).
7. Tang ZZ, *et al.* Multi-Omic Analysis of the Microbiome and Metabolome in Healthy Subjects Reveals Microbiome-Dependent Relationships Between Diet and Metabolites. *Front Genet* **10**, 454 (2019).
8. Zeng X, *et al.* Gut bacterial nutrient preferences quantified in vivo. *Cell* **185**,

3441-3456.e3419 (2022).

9. Zhao L, *et al.* Pivotal interplays between fecal metabolome and gut microbiome reveal functional signatures in cerebral ischemic stroke. *J Transl Med* **20**, 459 (2022).
10. Diener C, *et al.* Genome-microbiome interplay provides insight into the determinants of the human blood metabolome. *Nat Metab*, (2022).
11. Lofffield E, *et al.* Association of body mass index with fecal microbial diversity and metabolites in the northern Finland birth cohort. *Cancer Epidemiology, Biomarkers & Prevention* **29**, 2289-2299 (2020).
12. Cui M, *et al.* Human fecal metabolome reflects differences in body mass index, physical fitness, and blood lipoproteins in healthy older adults. *Metabolites* **11**, 717 (2021).
13. Carayol M, *et al.* Blood Metabolic Signatures of Body Mass Index: A Targeted Metabolomics Study in the EPIC Cohort. *J Proteome Res* **16**, 3137-3146 (2017).
14. Moore SC, *et al.* Human metabolic correlates of body mass index. *Metabolomics* **10**, 259-269 (2014).
15. Jia H, *et al.* Metabolomic analyses reveal new stage-specific features of COVID-19. *European Respiratory Journal* **59**, (2022).
16. Liu T, *et al.* Aberrant amino acid metabolism promotes neurovascular reactivity in rosacea. *JCI Insight*, (2022).
17. Jiang Y, *et al.* Plasma metabolomics of schizophrenia with cognitive impairment: A pilot study. *Frontiers in psychiatry* **13**, (2022).
18. Pang Y, *et al.* Associations of Adiposity, Circulating Protein Biomarkers, and Risk of Major Vascular Diseases. *JAMA Cardiol* **6**, 276-286 (2021).
19. Deng K, *et al.* Temporal relationship among adiposity, gut microbiota, and insulin resistance in a longitudinal human cohort. *BMC Med* **20**, 171 (2022).
20. Chen L, *et al.* Gut microbial co-abundance networks show specificity in inflammatory bowel disease and obesity. *Nat Commun* **11**, 4018 (2020).
21. Qi J, *et al.* Circulating trimethylamine N-oxide and the risk of cardiovascular diseases: a systematic review and meta-analysis of 11 prospective cohort studies. *Journal of cellular and molecular medicine* **22**, 185-194 (2018).

22. Lyu Y, *et al.* Associations between dietary patterns and physical fitness among Chinese elderly. *Public Health Nutrition* **24**, 4466-4473 (2021).
23. Qin B, Plassman BL, Edwards LJ, Popkin BM, Adair LS, Mendez MA. Fish intake is associated with slower cognitive decline in Chinese older adults. *The Journal of nutrition* **144**, 1579-1585 (2014).

REVIEWERS' COMMENTS

Reviewer #1 (Remarks to the Author):

The authors have addressed most of our critiques in a satisfactory manner. We believe, however, that a comprehensive language editing is still required, and detail our remaining concerns below.

- Language editing is still needed to improve the grammar and conciseness of the text. This is especially crucial for the abstract and discussion. A few examples are listed below:

o Line 38: "Most paired fecal and blood metabolites had low correlations in the Spearman correlation analysis."  "We found that most paired fecal and blood metabolites were not significantly correlated".

o Line 52: The human gut microbiome is a... (missing "The").

o Missing a point at the end of line 62

o Line 65-70: The sentence is currently phrased as following "While previous evidence demonstrated that..., other studies showed....". But these two parts are not conflicting and so the sentence structure is inappropriate...

o Line 70: "blood was the most widely used". This statement is not justified, say widely used without the "most".

o Line 80: "Currently, the majority of existing gut microbiota-metabolome association studies". Delete "existing"

o Line 94: "...with multi-omics datasets of gut metagenomic sequencing (using fecal samples), and targeted quantitative fecal and blood metabolome collected at the same time point." (missing "and")

o Line 111: "After excluding metabolites with the proportion of the missing value ≥ 0.2 or with..."  "After excluding metabolites with over 20% missing values or with..."

o Line 126: "using the partial Spearman correlation analysis". Delete "the" and "analysis"

o Line 145: "were overlapped". Delete "were"

o Line 163: "hold-out"  "held-out"

o Line 212-218 requires complete rephrasing. Totally unclear.

o Line 233-234 – unclear.

o Line 267: "which were used" - where?

This is a very partial list – the entire text requires similar corrections.

- The paragraph starting at line 198 is unclear. It seems like you are reporting the same result twice. (i.e. that fecal-metabolite models outperform their corresponding blood-metabolite models).

- The description of figure 3 is confusing. For example, in fig. 3D: “well-predicted metabolites that are only identified in the blood” – you meant “metabolites that were only well-predicted in the blood and not in feces”, right?

- The use of heterogeneity tests (with a cutoff of $P > 0.05$) to state that a well-predicted metabolite is validated (either in another cohort or in the other specimen type) is problematic and should either be revised or better explained/justified in the methods (lines 212-218, 524-527, 884-886,). If a model showed decent performance (e.g. $\text{spearman} > 0.3$) in both discovery and validation cohorts, it is obviously validated. We understand that the motivation for using heterogeneity tests was to be able to call additional metabolites “validated”, but if they do not pass the spearman threshold the evidence is weaker and this point should be clarified.

- Font size in some plots is still very small (e.g. Fig3D)

Reviewer #2 (Remarks to the Author):

The manuscript has improved in the revision, and the authors have met most of the comments in the previous round. However there is still some issues relating to language and the validation.

1. The language is still somewhat poor and difficult to understand. Here are a few examples of sentences that are not clear to me:

“To investigate and compare the roles of fecal and blood metabolites in linking the gut microbiota and cardiometabolic diseases”

“In summary, fecal metabolites, dominating over their blood counterpart, may provide a functional link between gut microbiota and cardiometabolic diseases”

“These findings were interesting for well-predicted metabolites that were identified in both fecal and blood samples”

2. I am not sure "blood microbiome" is a correct expression. In this paper it is the serum metabolome that is analyzed, which is quite different.
3. It needs to be clear in the abstract what number of metabolites and species that were analyzed and that the metabolite analysis was targeted.
4. It needs to be clear in abstract that prevalent cardiometabolic disease were assessed.
5. "the prevalence of cardiometabolic diseases" change to "prevalent cardiometabolic diseases". Here it is also questionable if obesity is a disease?
6. The response to the question around validation of the metabolite CVD results was not satisfactory. If the power was so low that replication could not be assessed with the same criteria as in the discovery cohort, I think that the power of the heterogeneity test might be even lower. Looking at the 4C it is evident that the CIs in the replication covers a large range of values, also those with opposite direction. I think that the claim that the "majority of our identified associations between well-predicted fecal metabolites and cardiometabolic diseases were robust" given the data provided.

Author's Response to Reviewer Comments

REVIEWERS' COMMENTS

Reviewer #1 (Remarks to the Author):

The authors have addressed most of our critiques in a satisfactory manner. We believe, however, that a comprehensive language editing is still required, and detail our remaining concerns below.

- Language editing is still needed to improve the grammar and conciseness of the text. This is especially crucial for the abstract and discussion. A few examples are listed below:

o Line 38: “Most paired fecal and blood metabolites had low correlations in the Spearman correlation analysis.”  “We found that most paired fecal and blood metabolites were not significantly correlated”.

o Line 52: The human gut microbiome is a... (missing “The”).

o Missing a point at the end of line 62

o Line 65-70: The sentence is currently phrased as following “While previous evidence demonstrated that..., other studies showed...”. But these two parts are not conflicting and so the sentence structure is inappropriate...

o Line 70: “blood was the most widely used”. This statement is not justified, say widely used without the “most”.

o Line 80: “Currently, the majority of existing gut microbiota-metabolome association studies”. Delete "existing"

o Line 94: “...with multi-omics datasets of gut metagenomic sequencing (using fecal samples), and targeted quantitative fecal and blood metabolome collected at the same time point.” (missing “and”)

o Line 111: “After excluding metabolites with the proportion of the missing value ≥ 0.2 or with...”  “After excluding metabolites with over 20% missing values or with...”

o Line 126: “using the partial Spearman correlation analysis”. Delete "the" and "analysis"

o Line 145: “were overlapped”. Delete "were"

o Line 163: “hold-out”  “held-out”

o Line 212-218 requires complete rephrasing. Totally unclear.

o Line 233-234 – unclear.

o Line 267: “which were used” - where?

This is a very partial list – the entire text requires similar corrections.

Response: Thanks for your very helpful language editing suggestions. We have now revised the related sentences in the revised manuscript:

Lines 39-40: “We found that most paired fecal and blood metabolites were not significantly correlated.”

Line 54: “The human gut microbiome is a diverse and complex ecosystem”

Line 61-62: “substantially influencing the development of cardiometabolic diseases^{1, 3, 4}.”

Lines 65-69: “Previous evidence demonstrated that gut microbiota significantly explained the variance of blood metabolome (average explained variance: 4.6%~15%)^{5, 6, 7, 8}, and meanwhile, other studies showed that the gut microbial composition was largely reflected by fecal metabolome (average explained variance: 67.7%)”

Lines 70-72: “In epidemiological studies, blood was the widely used biological sample for metabolome profiling to facilitate the exploration of the association between gut microbiota-related metabolites and cardiometabolic diseases”

Lines 80-81: “Currently, the majority of gut microbiota-metabolome association studies were performed using either feces or blood for metabolome profiling”

Lines 94-96: “with multi-omics datasets of gut metagenomic sequencing (using fecal samples), and targeted quantitative fecal and blood metabolome collected at the same time point”

Lines 112-114: “After excluding metabolites with over 20% missing values or with relative standard deviation (standard deviation/mean) ≥ 0.3 in quality control samples”

Lines 124-126: “We first assessed the phenotypic correlations (a direct comparison of each metabolite’s values between feces and blood) between paired fecal and blood metabolites using partial *Spearman* correlation adjusted for age, sex, and BMI.”

Lines 143-145: “Among them, three metabolites (3-indolepropionic acid, glycocholic acid, and phenylacetic acid) overlapped with those of phenotypic correlations (Fig. 2b).”

Lines 161-163: “*Spearman*’s correlation between the measured and predicted metabolite levels for held-out samples by machine learning pipeline”

Line 497: “with 202, 202, 201, 201, 201 held-out samples, respectively”

Lines 498-500: “We calculated *Spearman*’s correlation coefficient r and *Spearman*’s correlation P value between the measured and predicted metabolite levels for held-out samples.”

Lines 207-213: “Based on *Spearman*’s correlation coefficient > 0.3 and FDR < 0.05 (Methods), 68% (132/194) associations could be validated, including 63.3% (62/98)

taxonomic composition-fecal metabolite associations, 80% (8/10) taxonomic composition-blood metabolite associations, 71.4% (55/77) microbial pathways-fecal metabolite associations, and 77.8% (7/9) microbial pathways-blood metabolite associations (Fig. 3e, Supplementary Fig. 4e and Supplementary Fig. 9 and Supplementary Data 4).”

Lines 228-229: “These findings were interesting for metabolites that were well-predicted in both blood and feces”

Lines 263-264: “We further showed that some blood metabolites were weakly associated with gut microbiota or cardiometabolic diseases.”

Besides revising the sentences listed by the reviewer, we have made additional language editing to improve the grammar and conciseness:

In the revised manuscript:

Lines 32-33: “Blood metabolome has been widely used in clinical studies to explore the associations of gut microbiota-derived metabolites with cardiometabolic diseases”

Lines 73-75: “Nemet *et al.* found that blood phenylacetylglutamine was positively associated with incident cardiovascular disease and major adverse cardiovascular events”

Lines 75-77: “Vangipurapu *et al.* identified several gut microbiota-related blood metabolites including kynurenate, dimethylglycine, 2-hydroxyhippurate that were associated with the risk of type 2 diabetes”

Lines 83-85: “fecal and blood samples were not collected at the same time points and were obtained from different subsets of participants”

Lines 128-129: “There were only eight significantly correlated metabolites between feces and blood”

Lines 149-152: “Given that there were low correlations between most paired fecal and blood metabolites, we explored whether there were differences between the associations of taxonomic composition or microbial pathways with paired fecal and blood metabolites”

Lines 171-172: “We then considered metabolites with correlation coefficient > 0.3 and FDR < 0.05 as well-predicted metabolites”

Lines 180-182: “There were 10 well-predicted blood metabolites (Fig. 3b, Fig. 3e and Supplementary Data 2), among which, 8 overlapped with fecal samples”

Lines 198-199: “the number of well-predicted fecal metabolites was substantially more than that of blood metabolites”

Lines 242-243: “Comparison between paired fecal and blood SCFAs in associating gut microbiota and T2D”

Lines 244-245: “SCFAs, such as acetic acid, propionic acid, and butyric acid are well-known gut microbiota-derived metabolites”

Line 254-255: “we found that fecal butyric acid levels were inversely associated with prevalent T2D”

Lines 260-261: “we compared paired fecal and blood metabolome in their associations with gut microbiota and cardiometabolic diseases”

Lines 296-297: “The fecal butyric acid levels were inversely (beneficially) associated with T2D in our present study, which has been demonstrated by several animal models”

Lines 298-299: “In human studies, higher fecal butyric acid levels were associated with improved insulin response”

Lines 301-303: “We found that fecal hydrocinnamic acid levels were inversely associated with T2D, which was consistent with Vangipurapu et al.’s study, where blood levels of this metabolite were measured”

Lines 325-326: “Second, there are only 132 metabolites analyzed in our study because of the targeted metabolome profiling”

Lines 336-337: “the corresponding associations are unable to be validated.”

Lines 338-339: “as the participants enrolled in our study are not selected at random, there is potential selection bias.”

Lines 476-477: “The detailed information for the genotyping data was described in our previous paper”

Lines 479-481: “Phenotypic or genetic correlations between paired fecal and blood metabolites with the absolute value of correlation coefficients $|r| > 0.3$ and $FDR < 0.05$ were considered significant.”

Lines 491-492: “As RF and LightGBM models were commonly used in the gut microbiota-metabolome association studies”

Lines 516: “the main results were reported based on the RF method throughout this study.”

Lines 523-524: “Independent validation for the identified gut microbiota-fecal/blood metabolite associations”

Lines 525-526: “We then attempted to replicate the above identified associations in an independent validation cohort”

Lines 547-548: “*Pearson* correlation between the partial regression coefficients obtained from the discovery and validation cohort was calculated”

Lines 821-824: “A total of 1007 participants from the Guangzhou Nutrition and Health Study with matched gut metagenomic and fecal and blood metabolomics data and without taking antibiotics within two weeks are included in this study.”

Lines 827-828: “After removing metabolites with missing rate ≥ 0.2 or with relative standard deviation ≥ 0.3 ”

Lines 829-832: “We estimate the gut microbiota-fecal/blood metabolite associations using the machine learning pipeline, and compare the associations of taxonomic composition/microbial pathways with paired fecal and blood metabolites.”

Lines 843-844: “Correlations with $FDR < 0.05$ and $|r| > 0.3$ (red dashed lines) are considered significant”

Lines 846-848: “For genetic correlation analysis, there are 38 metabolites that do not converged during the calculation process and thus have no results.”

Lines 860-862: “The difference between the distributions of taxonomic composition-fecal metabolite associations and taxonomic composition-blood metabolite associations is tested by Wilcoxon signed-rank test”

Lines 879-881: “Associations with *Spearman*'s correlation coefficient > 0.3 and $FDR < 0.05$ are considered as being validated in the validation cohort.”

In the revised supplementary notes:

Line 8 and line 46: “Performance comparison between the RF and LightGBM model”

Lines 49-51: “we used both of them to predict the concentrations of fecal/blood metabolites based on the taxonomic composition/microbial pathways and compared their performances”

Lines 51-53: “We first computed the root mean square error (RMSE) for each model that predict each metabolite levels based on taxonomic composition or microbial pathways.”

Lines 55-56: “Supplementary Fig. 1b showed that there was a strong correlation between the predictability results of metabolites obtained by the RF and LightGBM model”

Lines 67-68: “while there was only one additional association that was obtained using the LightGBM model”

In the revised supplementary figures:

Lines 100-102: “The correlation between the predictability results of metabolites obtained by the RF and LightGBM model is computed by *Pearson* correlation.”

Lines 127-128: “Correlations with $FDR < 0.05$ and $|r| > 0.3$ (red dashed lines) are considered significant”

Lines 139-140: “Correlation between phenotypic correlations with and without additional adjustment for BMI is calculated by *Pearson* correlation.”

Lines 155-157: “The difference between the distributions of microbial pathways-fecal metabolite associations and microbial pathways-blood metabolite associations is tested by Wilcoxon signed-rank test.”

Lines 174-176: “Associations with *Spearman*'s correlation coefficient > 0.3 and $FDR < 0.05$ are considered as being validated in the validation cohort.”

Lines 180-182: “The overlap between well-predicted metabolites based on taxonomic composition and microbial pathways for fecal metabolites and (b) for blood metabolites.”

Lines 233-234: “Sensitivity analysis for the identified associations between well-predicted fecal metabolites and prevalent cardiometabolic diseases”

- The paragraph starting at line 198 is unclear. It seems like you are reporting the same result twice. (i.e. that fecal-metabolite models outperform their corresponding blood-metabolite models).

Response: We have now revised this paragraph and integrated it with the previous paragraph (lines 187-196).

Lines 187-196: “Most well-predicted fecal metabolites had superior predictability over their corresponding paired blood metabolites, and for metabolites that were well-predicted in both feces and blood, the models of most well-predicted fecal metabolites significantly outperformed their respective models of well-predicted blood metabolites (FDR < 0.05, see Methods; Fig. 3d, Supplementary Fig. 7a and Supplementary Data 3). The identified well-predicted metabolites based on microbial pathways showed consistent results, in which all identified well-predicted metabolites based on microbial pathways belonged to the identified well-predicted metabolites based on taxonomic composition (Supplementary Fig. 4, Supplementary Fig. 5a and Supplementary Fig. 5b and Supplementary Fig. 7b; Supplementary Data 2-3).”

- The description of figure 3 is confusing. For example, in fig. 3D: “well-predicted metabolites that are only identified in the blood” – you meant “metabolites that were only well-predicted in the blood and not in feces”, right?

Response: Sorry for our unclear description. We have now revised the related sentences in the revised manuscript (lines 865-869, lines 873-875, lines 184-187) and revised supplementary figures (lines 160-164, lines 168-169, lines 206-212).

In the revised manuscript:

Lines 865-869: “Differences between the associations of taxonomic composition with paired fecal and blood metabolites that are well-predicted in both feces and blood (marked with red in y axis), metabolites that are only well-predicted in blood and not in feces (marked with green in y axis), and the top-30 metabolites that are only well-predicted in feces and not in blood (marked with blue in y axis).”

Lines 873-875: “The results for the top 31-90 metabolites that are only well-predicted in feces and not in blood are presented in Supplementary Fig. 7a”

Lines 184-187: “There were 90 metabolites that were only well-predicted in feces and not in blood, and 2 metabolites (alpha-N-Phenylacetyl-L-glutamine and hippuric acid) that were only well-predicted in blood and not in feces.”

In the revised supplementary figures:

Lines 160-164: “Differences between the associations of microbial pathways with paired fecal and blood metabolites that are well-predicted in both feces and blood (marked with red in y axis), metabolites that are only well-predicted in blood and not

in feces (marked with green in y axis), and the top-30 metabolites that are only well-predicted in feces and not in blood (marked with blue in y axis).”

Lines 168-169: “The results for the top 31-70 metabolites that are only well-predicted in feces and not in blood are presented in Supplementary Fig. 7b”

Lines 206-212: (a) Differences between the associations of taxonomic composition with paired fecal and blood metabolites for top 31-90 metabolites that are only well-predicted in feces and not in blood and are ranked by the predictability of fecal metabolites. (b) Differences between the associations of microbial pathways with paired fecal and blood metabolites for top 31-70 metabolites that are only well-predicted in feces and not in blood and are ranked by the predictability of fecal metabolites.”

- The use of heterogeneity tests (with a cutoff of $P > 0.05$) to state that a well-predicted metabolite is validated (either in another cohort or in the other specimen type) is problematic and should either be revised or better explained/justified in the methods (lines 212-218, 524-527, 884-886, ...). If a model showed decent performance (e.g. $\text{spearman} > 0.3$) in both discovery and validation cohorts, it is obviously validated. We understand that the motivation for using heterogeneity tests was to be able to call additional metabolites “validated”, but if they do not pass the spearman threshold the evidence is weaker and this point should be clarified.

Response:

(1) Thanks for your advice. As suggested, we have now adopted the *Spearman*’s correlation > 0.3 and $\text{FDR} < 0.05$ as the criteria to validate gut microbiota-metabolite associations in the validation cohort. Based on this criteria, 68% (132/194) associations could be validated, including 63.3% (62/98) taxonomic composition-fecal metabolite associations, 80% (8/10) taxonomic composition-blood metabolite associations, 71.4% (55/77) microbial pathways-fecal metabolite associations, and 77.8% (7/9) microbial pathways-blood metabolite associations. We have revised the related sentences in the revised manuscript (lines 207-212, lines 531-532, and lines 879-881) and revised supplementary figures (lines 174-176, lines 258-259). In addition, the corresponding Supplementary Data 4, Fig. 3e, Supplementary Fig. 4e and Supplementary Fig. 9, and Source Data were also updated.

In the revised manuscript:

Lines 207-212: “Based on *Spearman*’s correlation coefficient > 0.3 and $\text{FDR} < 0.05$ (Methods), 68% (132/194) associations could be validated, including 63.3% (62/98) taxonomic composition-fecal metabolite associations, 80% (8/10) taxonomic composition-blood metabolite associations, 71.4% (55/77) microbial pathways-fecal

metabolite associations, and 77.8% (7/9) microbial pathways-blood metabolite associations”

Lines 531-532: “We considered the associations with *Spearman*’s correlation coefficient > 0.3 and $FDR < 0.05$ as being validated.”

Lines 879-881: “Associations with *Spearman*’s correlation coefficient > 0.3 and $FDR < 0.05$ are considered as being validated in the validation cohort.”

In the revised supplementary figures:

Lines 174-176: “Associations with *Spearman*’s correlation coefficient > 0.3 and $FDR < 0.05$ are considered as being validated in the validation cohort.”

Lines 258-259: “Associations with *Spearman*’s correlation coefficient > 0.3 and $FDR < 0.05$ are considered as being validated in the validation cohort.”

(2) Instead of using heterogeneity tests, we now used an alternative method proposed by Hittner *et al.* to test the difference between associations of taxonomic composition/microbial pathways with paired fecal and blood metabolites for each well-predicted metabolite (i.e. comparing two correlation coefficients) by R package cocor (version: 1.1-4)^{1, 2}, and obtained more significant results (Fig. 3d and Supplementary Fig. 4d and Supplementary Fig. 7). We have revised the related sentences in the revised manuscript (lines 518-521, lines 189-192 and lines 865-866) and revised supplementary figures (lines 160-161, lines 205-216). In addition, the corresponding Fig. 3d, Supplementary Fig. 4d and Supplementary Fig. 7, Supplementary Data 3, and Source Data were updated.

In the revised manuscript:

Lines 518-521: “We assessed the difference between the associations of taxonomic composition/microbial pathways with paired fecal and blood metabolites for each well-predicted metabolite using the method proposed by Hittner *et al.*⁵⁴, which was implemented by R package cocor (version: 1.1-4)⁵⁵.”

Lines 189-192: “the models of most well-predicted fecal metabolites significantly outperformed their respective models of well-predicted blood metabolites ($FDR < 0.05$, see Methods; Fig. 3d, Supplementary Fig. 7a and Supplementary Data 3).”

Lines 865-866: “Differences between the associations of taxonomic composition with paired fecal and blood metabolites”

In the revised supplementary figures:

Lines 160-161: “Differences between the associations of microbial pathways with paired fecal and blood metabolites that are well-predicted in both feces and blood”

Lines 205-216: “Supplementary Fig. 7. Differences between the associations of gut microbiota with paired fecal and blood metabolites. (a) Differences between the associations of taxonomic composition with paired fecal and blood metabolites for top 31-90 metabolites that are only well-predicted in feces and not in blood and are ranked by the predictability of fecal metabolites. (b) Differences between the associations of microbial pathways with paired fecal and blood metabolites for top 31-70 metabolites that are only well-predicted in feces and not in blood and are ranked by the predictability of fecal metabolites. Differences between the associations of taxonomic composition/microbial pathways with paired fecal and blood metabolites are tested by the method proposed by Hittner et al. (see Methods). FDR is controlled by the Benjamini-Hochberg method. *FDR < 0.05, **FDR < 0.01, *** FDR < 0.005. All statistical tests are two-sided. Source data are provided as a Source Data file.”

- Font size in some plots is still very small (e.g. Fig3D)

Response: We have now increased the font size in Fig. 3 and Supplementary Fig. 4.

Reviewer #2 (Remarks to the Author):

The manuscript has improved in the revision, and the authors have met most of the comments in the previous round. However there is still some issues relating to language and the validation.

1. The language is still somewhat poor and difficult to understand. Here are a few examples of sentences that are not clear to me:

“To investigate and compare the roles of fecal and blood metabolites in linking the gut microbiota and cardiometabolic diseases”

“In summary, fecal metabolites, dominating over their blood counterpart, may provide a functional link between gut microbiota and cardiometabolic diseases”

“These findings were interesting for well-predicted metabolites that were identified in both fecal and blood samples”

Response: Thanks for your comments. We have revised the related sentences in the revised manuscript:

Lines 33-35: “To systematically compare the roles of fecal and blood metabolites in linking the gut microbiota and cardiometabolic diseases”

Lines 46-48: “In summary, fecal metabolites, compared with their blood counterpart, may be more relevant in providing a functional link between gut microbiota and cardiometabolic diseases.”

Lines 228-229: “These findings were interesting for metabolites that were well-predicted in both blood and feces”

Besides revising the sentences listed by two reviewers, we have made additional language editing to improve the manuscript:

In the revised manuscript:

Lines 32-33: “Blood metabolome has been widely used in clinical studies to explore the associations of gut microbiota-derived metabolites with cardiometabolic diseases”

Lines 73-75: “Nemet *et al.* found that blood phenylacetylglutamine was positively associated with incident cardiovascular disease and major adverse cardiovascular events”

Lines 75-77: “Vangipurapu et al. identified several gut microbiota-related blood metabolites including kynurenate, dimethylglycine, 2-hydroxyhippurate that were associated with the risk of type 2 diabetes”

Lines 83-85: “fecal and blood samples were not collected at the same time points and were obtained from different subsets of participants”

Lines 128-129: “There were only eight significantly correlated metabolites between feces and blood”

Lines 149-152: “Given that there were low correlations between most paired fecal and blood metabolites, we explored whether there were differences between the associations of taxonomic composition or microbial pathways with paired fecal and blood metabolites”

Lines 171-172: “We then considered metabolites with correlation coefficient > 0.3 and FDR < 0.05 as well-predicted metabolites”

Lines 180-182: “There were 10 well-predicted blood metabolites (Fig. 3b, Fig. 3e and Supplementary Data 2), among which, 8 overlapped with fecal samples”

Lines 198-199: “the number of well-predicted fecal metabolites was substantially more than that of blood metabolites”

Lines 242-243: “Comparison between paired fecal and blood SCFAs in associating gut microbiota and T2D”

Lines 244-245: “SCFAs, such as acetic acid, propionic acid, and butyric acid are well-known gut microbiota-derived metabolites”

Line 254-255: “we found that fecal butyric acid levels were inversely associated with prevalent T2D”

Lines 260-261: “we compared paired fecal and blood metabolome in their associations with gut microbiota and cardiometabolic diseases”

Lines 296-297: “The fecal butyric acid levels were inversely (beneficially) associated with T2D in our present study, which has been demonstrated by several animal models”

Lines 298-299: “In human studies, higher fecal butyric acid levels were associated with improved insulin response”

Lines 301-303: “We found that fecal hydrocinnamic acid levels were inversely associated with T2D, which was consistent with Vangipurapu et al.’s study, where blood levels of this metabolite were measured”

Lines 325-326: “Second, there are only 132 metabolites analyzed in our study because of the targeted metabolome profiling”

Lines 336-337: “the corresponding associations are unable to be validated.”

Lines 338-339: “as the participants enrolled in our study are not selected at random, there is potential selection bias.”

Lines 476-477: “The detailed information for the genotyping data was described in our previous paper”

Lines 479-481: “Phenotypic or genetic correlations between paired fecal and blood metabolites with the absolute value of correlation coefficients $|r| > 0.3$ and FDR < 0.05 were considered significant.”

Lines 491-492: “As RF and LightGBM models were commonly used in the gut microbiota-metabolome association studies”

Lines 516: “the main results were reported based on the RF method throughout this study.”

Lines 523-524: “Independent validation for the identified gut microbiota-fecal/blood metabolite associations”

Lines 525-526: “We then attempted to replicate the above identified associations in an independent validation cohort”

Lines 547-548: “*Pearson* correlation between the partial regression coefficients obtained from the discovery and validation cohort was calculated”

Lines 821-824: “A total of 1007 participants from the Guangzhou Nutrition and Health Study with matched gut metagenomic and fecal and blood metabolomics data and without taking antibiotics within two weeks are included in this study.”

Lines 827-828: “After removing metabolites with missing rate ≥ 0.2 or with relative standard deviation ≥ 0.3 ”

Lines 829-832: “We estimate the gut microbiota-fecal/blood metabolite associations using the machine learning pipeline, and compare the associations of taxonomic composition/microbial pathways with paired fecal and blood metabolites.”

Lines 843-844: “Correlations with $FDR < 0.05$ and $|r| > 0.3$ (red dashed lines) are considered significant”

Lines 846-848: “For genetic correlation analysis, there are 38 metabolites that do not converged during the calculation process and thus have no results.”

Lines 860-862: “The difference between the distributions of taxonomic composition-fecal metabolite associations and taxonomic composition-blood metabolite associations is tested by Wilcoxon signed-rank test”

Lines 879-881: “Associations with *Spearman*'s correlation coefficient > 0.3 and $FDR < 0.05$ are considered as being validated in the validation cohort.”

In the revised supplementary notes:

Line 8 and line 46: “Performance comparison between the RF and LightGBM model”

Lines 49-51: “we used both of them to predict the concentrations of fecal/blood metabolites based on the taxonomic composition/microbial pathways and compared their performances”

Lines 51-53: “We first computed the root mean square error (RMSE) for each model

that predict each metabolite levels based on taxonomic composition or microbial pathways.”

Lines 55-56: “Supplementary Fig. 1b showed that there was a strong correlation between the predictability results of metabolites obtained by the RF and LightGBM model”

Lines 67-68: “while there was only one additional association that was obtained using the LightGBM model”

In the revised supplementary figures:

Lines 100-102: “The correlation between the predictability results of metabolites obtained by the RF and LightGBM model is computed by *Pearson* correlation.”

Lines 127-128: “Correlations with $FDR < 0.05$ and $|r| > 0.3$ (red dashed lines) are considered significant”

Lines 139-140: “Correlation between phenotypic correlations with and without additional adjustment for BMI is calculated by *Pearson* correlation.”

Lines 155-157: “The difference between the distributions of microbial pathways-fecal metabolite associations and microbial pathways-blood metabolite associations is tested by Wilcoxon signed-rank test.”

Lines 174-176: “Associations with *Spearman*’s correlation coefficient > 0.3 and $FDR < 0.05$ are considered as being validated in the validation cohort.”

Lines 180-182: “The overlap between well-predicted metabolites based on taxonomic composition and microbial pathways for fecal metabolites and (b) for blood metabolites.”

Lines 233-234: “Sensitivity analysis for the identified associations between well-predicted fecal metabolites and prevalent cardiometabolic diseases”

2. I am not sure ”blood microbiome” is a correct expression. In this paper it is the serum metabolome that is analyzed, which is quite different.

Response: We suspect what this reviewer mentioned is “blood metabolome”. We think the “blood metabolome” is a correct expression, given that it has been used in many high-quality and high-impact studies^{3, 4, 5, 6, 7, 8}.

3. It needs to be clear in the abstract what number of metabolites and species that were

analyzed and that the metabolite analysis was targeted.

Response: Thanks for your suggestion. We have now added the relevant information in the revised manuscript (lines 35-39). We have also added the number of species and pathways in the methods section of the revised manuscript (lines 407-409).

Lines 35-39: “we performed comparisons between paired fecal and blood metabolites in their associations with gut microbiota and cardiometabolic diseases among 1007 middle-aged and elderly adults with matched gut metagenomic (149 species and 214 pathways), and fecal and blood targeted metabolomics data (132 metabolites).”

Lines 407-409: “We included microbial species and pathways with a minimum detective relative abundance of 0.01% in at least 10% of the samples, which yielded 149 species and 214 pathways.”

4. It needs to be clear in abstract that prevalent cardiometabolic disease were assessed.

Response: Done (lines 42-44)

Lines 42-44: “The blood metabolites had a weak role in associating gut microbiota and prevalent cardiometabolic diseases. For example, fecal, but not blood butyric acid was associated with both gut microbiota and prevalent type 2 diabetes.”

5. ”the prevalence of cardiometabolic diseases” change to ”prevalent cardiometabolic diseases”. Here it is also questionable if obesity is a disease?

Response: We have changed “the prevalence of cardiometabolic diseases” to “prevalent cardiometabolic diseases” (lines 216-218, lines 536-537, lines 254-255). In addition, according to the position statement of the Obesity Society, obesity is a noncommunicable chronic disease^{9, 10}.

Lines 216-218: “we explored the associations of the above well-predicted fecal and blood metabolites identified in the discovery cohort with prevalent cardiometabolic diseases”

Lines 536-537: “We examined the associations of well-predicted fecal and blood metabolites with prevalent cardiometabolic diseases”

Lines 254-255: “Furthermore, we found that fecal butyric acid levels were inversely associated with prevalent T2D”

6. The response to the question around validation of the metabolite CVD results was

not satisfactory. If the power was so low that replication could not be assessed with the same criteria as in the discovery cohort, I think that the power of the heterogeneity test might be even lower. Looking at the 4C it is evident that the CIs in the replication covers a large range of values, also those with opposite direction. I think that the claim that the “majority of our identified associations between well-predicted fecal metabolites and cardiometabolic diseases were robust” given the data provided.

Response: Thanks for your comments. We now abandoned the use of heterogeneity test in this new version of manuscript. We have revised the related sentences in the revised manuscript (lines 544-547, lines 893-894, and 232-239). In addition, the corresponding Fig. 4c and Source Data were also updated.

Lines 545-548: “The results were further replicated in the validation cohort. Only T2D ($N_{\text{patients}} = 27$), hypertension ($N_{\text{patients}} = 23$), and obesity ($N_{\text{patients}} = 14$) were available in the validation cohort. *Pearson* correlation between the partial regression coefficients obtained from the discovery and validation cohort was calculated.”

Lines 893-894: “(c) Replication of the significant associations of well-predicted fecal metabolites with cardiometabolic diseases in the validation cohort.”

Lines 233-240: “The low replication rate shown in Fig. 4c was largely a power issue as the sample size of the validation cohort was small. However, we discovered that 8 out of 11 associations for the well-predicted fecal metabolites had the same effect directions and strong correlation between partial regression coefficients obtained from the discovery and validation cohort ($r = 0.894$; Fig. 4cd), which suggested that the majority of our identified associations between well-predicted fecal metabolites and cardiometabolic diseases were consistent across the discovery and validation cohort.”

Reference

1. Hittner JB, May K, Silver NC. A Monte Carlo evaluation of tests for comparing dependent correlations. *The Journal of general psychology* **130**, 149-168 (2003).
2. Diedenhofen B, Musch J. cocor: A comprehensive solution for the statistical comparison of correlations. *PloS one* **10**, e0121945 (2015).
3. Wilmanski T, *et al.* Blood metabolome predicts gut microbiome α -diversity in humans. *Nature biotechnology* **37**, 1217-1228 (2019).
4. Diener C, *et al.* Genome-microbiome interplay provides insight into the determinants of the human blood metabolome. *bioRxiv*, (2022).

5. Sun L, *et al.* Association between Human Blood Metabolome and the Risk of Alzheimer's Disease. *Ann Neurol* **92**, 756-767 (2022).
6. Li Z, *et al.* Evaluation of the Use of Saliva Metabolome as a Surrogate of Blood Metabolome in Assessing Internal Exposures to Traffic-Related Air Pollution. *Environ Sci Technol* **56**, 6525-6536 (2022).
7. Guida F, *et al.* The blood metabolome of incident kidney cancer: A case-control study nested within the MetKid consortium. *PLoS Med* **18**, e1003786 (2021).
8. Pinto J, *et al.* Impact of fetal chromosomal disorders on maternal blood metabolome: toward new biomarkers? *Am J Obstet Gynecol* **213**, 841.e841-841.e815 (2015).
9. Jastreboff AM, Kotz CM, Kahan S, Kelly AS, Heymsfield SB. Obesity as a Disease: The Obesity Society 2018 Position Statement. *Obesity (Silver Spring)* **27**, 7-9 (2019).
10. Obesity as a disease: the Obesity Society Council resolution. *Obesity (Silver Spring)* **16**, 1151 (2008).